# Hypothalamic astrocyte NAD+ salvage pathway mediates the coupling of dietary fat overconsumption in a mouse model of obesity

Jae Woo Park[1], Se Eun Park[1], Wuhyun Koh [2], Won Hee Jang [1], Jong Han Choi [3], Eun Roh [4], Gil Myoung Kang[5], Seong Jun Kim[1], Hyo Sun Lim[1], Chae Beom Park [1], So Yeon Jeong[1], Sang Yun Moon[1], Chan Hee Lee [6], Sang Yeob Kim[7], Hyung Jin Choi [8], Se Hee Min [5,9], C. Justin Lee [2] & Min-Seon Kim [5,9] ✉

Nicotinamide adenine dinucleotide (NAD)+ serves as a crucial coenzyme in numerous essential biological reactions, and its cellular availability relies on the activity of the nicotinamide phosphoribosyltransferase (NAMPT)-catalyzed salvage pathway. Here we show that treatment with saturated fatty acids activates the NAD+ salvage pathway in hypothalamic astrocytes. Furthermore, inhibition of this pathway mitigates hypothalamic inflammation and attenuates the development of obesity in male mice fed a high-fat diet (HFD). Mechanistically, CD38 functions downstream of the NAD+ salvage pathway in hypothalamic astrocytes burdened with excess fat. The activation of the astrocytic NAMPT–NAD+–CD38 axis in response to fat overload induces proinflammatory responses in the hypothalamus. It also leads to aberrantly activated basal Ca$^{2+}$ signals and compromised Ca$^{2+}$ responses to metabolic hormones such as insulin, leptin, and glucagon-like peptide 1, ultimately resulting in dysfunctional hypothalamic astrocytes. Our findings highlight the significant contribution of the hypothalamic astrocytic NAD+ salvage pathway, along with its downstream CD38, to HFD-induced obesity.

The recent global epidemic of obesity is strongly linked to excessive consumption of high-fat diets (HFDs). Repetitive exposure to dietary saturated fatty acids triggers a state of mild inflammation in multiple tissues, including the hypothalamus, an essential brain region governing energy balance and adiposity[1–3]. HFD-induced inflammation in the hypothalamus is now recognized as a pivotal factor contributing to hypothalamic dysfunction, which is intricately associated with the initiation and progression of obesity[1].

[1]Department of Biomedical Science, Asan Medical Institute of Convergence Science and Technology, University of Ulsan College of Medicine, Seoul 05505, Korea. [2]Center for Cognition and Sociality, Life Science Cluster, Institute for Basic Science, Daejeon 34126, Korea. [3]Division of Endocrinology and Metabolism, Konkuk University Medical Center, Seoul 05030, Korea. [4]Division of Endocrinology and Metabolism, Department of Internal Medicine, Hallym University Sacred Heart Hospital, Anyang 14068, Korea. [5]Appetite Regulation Laboratory, Asan Institute for Life Science, Seoul 05505, Korea. [6]Program of Material Science for Medicine and Pharmaceutics, Hallym University, Chuncheon 24252, Korea. [7]Asan Institute for Life Sciences, Asan Medical Center, Seoul 05505, Korea. [8]Department of Biomedical Sciences, Wide River Institute of Immunology, Seoul National University College of Medicine, Seoul 03080, Korea. [9]Division of Endocrinology and Metabolism, Asan Diabetes Center, Asan Medical Center, University of Ulsan College of Medicine, Seoul 05505, Korea. ✉e-mail: mskim@amc.seoul.kr

Emerging evidence indicates a potential contribution of hypothalamic non-neuronal cells glia in regulating body weight and peripheral nutrient metabolism[4,5]. Astrocytes, a prominent class of glial cells in the mammalian brain, play a vital role in supporting neuronal metabolism[6], modulating synaptic activity[7], and maintaining electrolyte and water balance[8]. In addition, astrocytes uphold the integrity of the blood-brain barrier[9] and regulate the transport of glucose into the brain[10].

Upon exposure to a HFD, astrocytes in the hypothalamic arcuate nucleus (ARH) undergo a transformation into a reactive state, characterized by increased expression of glial fibrillary acidic protein (GFAP), proinflammatory gene profiles, and hypertrophic cellular processes[11,12] – a phenomenon termed reactive astrogliosis[12]. The role of reactive hypothalamic astrocytes in the pathogenesis of obesity has gained significance. Artificial activation of nuclear factor-κB (NF-κB) signaling in astrocytes induces glucose intolerance, increased blood pressure, and adiposity in mice fed a normal diet[13,14]. Conversely, inhibiting NF-κB signaling suppresses HFD-induced obesity (DIO) and reactive astrogliosis and inflammation in the hypothalamus[13,14]. These findings underscore the role of astrocytic inflammatory signaling in the development of DIO. However, the intricate mechanisms through which hypothalamic astrocytes contribute to the progression of obesity remain elusive.

Nicotinamide adenine dinucleotide (NAD)$^+$ is an indispensable metabolite that serves as a cosubstrate in biochemical reactions catalyzed by sirtuins/SIRTs, poly(ADP-ribose) polymerase (PARP), and cyclic ADP-ribose cyclase/CD38[15,16]. NAD$^+$ is synthesized through three distinct precursor molecules and pathways: from tryptophan via de novo synthetic pathway, from nicotinic acid (NA) via the Press–Handler pathway, and from nicotinamide via the salvage pathway[17]. In mammals, the salvage pathway represents the primary means of maintaining cellular NAD$^+$ levels[15,16]. Through the NAD$^+$ salvage pathway, nicotinamide is converted back into NAD$^+$ via nicotinamide mononucleotide (NMN)[15,16]. A crucial step in the NAD$^+$ salvage pathway is catalyzed by nicotinamide phosphoribosyltransferase (NAMPT)[18].

Interestingly, NAMPT was initially identified as a proinflammatory cytokine capable of increasing the cellular expression of inflammatory cytokines and promoting pre-B cell colony formation[19,20]. Immune cells, including lymphocytes, dendritic cells, monocytes, and macrophages, upregulate NAMPT expression in response to inflammatory stimuli[21]. Moreover, FK866, a chemical inhibitor of NAMPT, has been shown to exhibit beneficial anti-inflammatory effects in various animal models of inflammation, including arthritis, acute lung injury, and spinal cord injury[22–24].

The NAD$^+$ salvage pathway has been implicated in various central nervous system (CNS) disorders, exhibiting either positive or negative effects depending on the specific pathological contexts or cell types[3,25,26]. The NAD$^+$–SIRT1 axis plays a pivotal role in regulating energy balance and circadian rhythms in the hypothalamic neurons[27–29]. Disruption of this pathway is intricately linked to the pathogenesis of aging and obesity[17,30,31]. However, the physiological and pathophysiological roles of NAD$^+$-relevant pathways in hypothalamic non-neuronal cells have yet to be explored.

In the current study, we have investigated the involvement of the astrocytic NAD$^+$ salvage pathway in regulating energy balance in the hypothalamus. Here, we demonstrate that the NAD$^+$ salvage pathway is activated in hypothalamic astrocytes during HFD feeding, thereby significantly contributing to the development of DIO.

## Results
### Saturated fat overload activates the NAD$^+$ salvage pathway in hypothalamic astrocytes
We first investigated whether cellular NAD$^+$ levels may be altered in different types of hypothalamic cells under fat-overloaded conditions.

To test this hypothesis, we conducted an in vitro experiment using N1 hypothalamic neuron cells and primary cultured hypothalamic astrocytes. These cells were exposed to saturated fatty acid palmitate at concentrations of 20 and 200 μM for varying durations (6, 24, and 48 h) to stimulate conditions of fat overload similar to those observed in obesity. Subsequently, we evaluated the cellular NAD$^+$ levels in response to these treatments.

After 6 h of palmitate treatment, the NAD$^+$ levels increased in hypothalamic neurons (Fig. 1a). However, at the 24- and 48-h time points, the NAD$^+$ levels in hypothalamic neurons decreased significantly (Fig. 1a), despite no alteration in cell viability (Supplementary Fig. 1). In contrast, exposure to palmitate for 24 and 48 h resulted in an increase in NAD$^+$ content in hypothalamic astrocytes, whereas no significant change was observed after 6 h of treatment (Fig. 1b). In addition, treatment with oleate, a monounsaturated fatty acid, led to no notable alterations in NAD$^+$ content in either cell type (Fig. 1a, b). Collectively, these findings suggest that an excessive intake of saturated fats may lead to distinct changes in NAD$^+$ levels in hypothalamic neurons and astrocytes.

We next conducted further analysis to investigate alterations in the NAD$^+$ biosynthetic pathway following palmitate treatment. Specifically, we examined the mRNA expression levels of enzymes associated with the NAD$^+$ salvage pathway (NAMPT, nicotinamide ribose kinase [NRK]−1,2), the Preiss–Handler pathway (nicotinic acid phosphoribosyltransferase [NaPRT]), the de novo synthetic pathway (quinolinic acid phosphoribosyltransferase [QaPRT]), and multiple pathways (nicotinamide mononucleotide adenylyltransferase [NMNAT]−1–3, NAD$^+$ synthase [NADSYN][16] (Fig. 1c). In hypothalamic neurons, 48-h treatment with palmitate downregulated the expression of Naprt and all subtypes of Nmnat, with no significant changes in the expression of other enzymes (Fig. 1d). This finding suggests that the suppression of Nmnat and Naprt expression by palmitate contributes to the reduction in hypothalamic neuronal NAD$^+$ levels. In hypothalamic astrocytes, palmitate treatment significantly increased the expression of Nampt and Nadsyn, suppressed Naprt expression, and had no significant effect on the expression of other NAD$^+$ biosynthetic enzymes (Fig. 1d). Consistent with this finding, exposure to palmitate also stimulated NAMPT protein levels and enzyme activity in hypothalamic astrocytes (Fig. 1e, f). Simultaneous treatment with the NAMPT chemical inhibitor FK866 or the genetic knockdown of Nampt mitigated the palmitate-induced increase in NAD$^+$ levels in hypothalamic astrocytes, with no significant effects on basal NAD$^+$ levels (Fig. 1g, h). This finding highlights the pivotal role of NAMPT in palmitate-induced elevation of NAD$^+$ levels in hypothalamic astrocytes. The interconversion between NAD$^+$ and NADH can influence cellular NAD$^+$ levels[16]; however, following 24- and 48-h palmitate treatment, no significant changes in astrocytic NADH levels were observed (Fig. 1i). Taken together, these findings imply that prolonged exposure to excessive dietary fat increases hypothalamic astrocyte NAD$^+$ levels through activation of the NAMPT-mediated NAD$^+$ salvage pathway.

To validate these findings in vivo, we examined the expression of Nampt in hypothalamic astrocytes of mice subjected to a 4-week HFD compared to those on a standard chow diet (CD) using Nampt fluorescence in situ hybridization (FISH) and GFAP double staining. In the CD-fed group, hypothalamic astrocytic Nampt expression was notably low, but it exhibited a significant increase in the HFD-fed group (Fig. 1j). Interestingly, no such changes were observed in astrocytes located in the hippocampus and cortex (Fig. 1j). Similarly, quantitative PCR (qPCR) analysis of Nampt in astrocytes isolated from different brain regions confirmed these findings, revealing increased Nampt expression in hypothalamic astrocytes due to HFD feeding, whereas cortical and hippocampal astrocytes showed no such increase (Fig. 1k). Consistently, the results of in vitro experiments demonstrated that

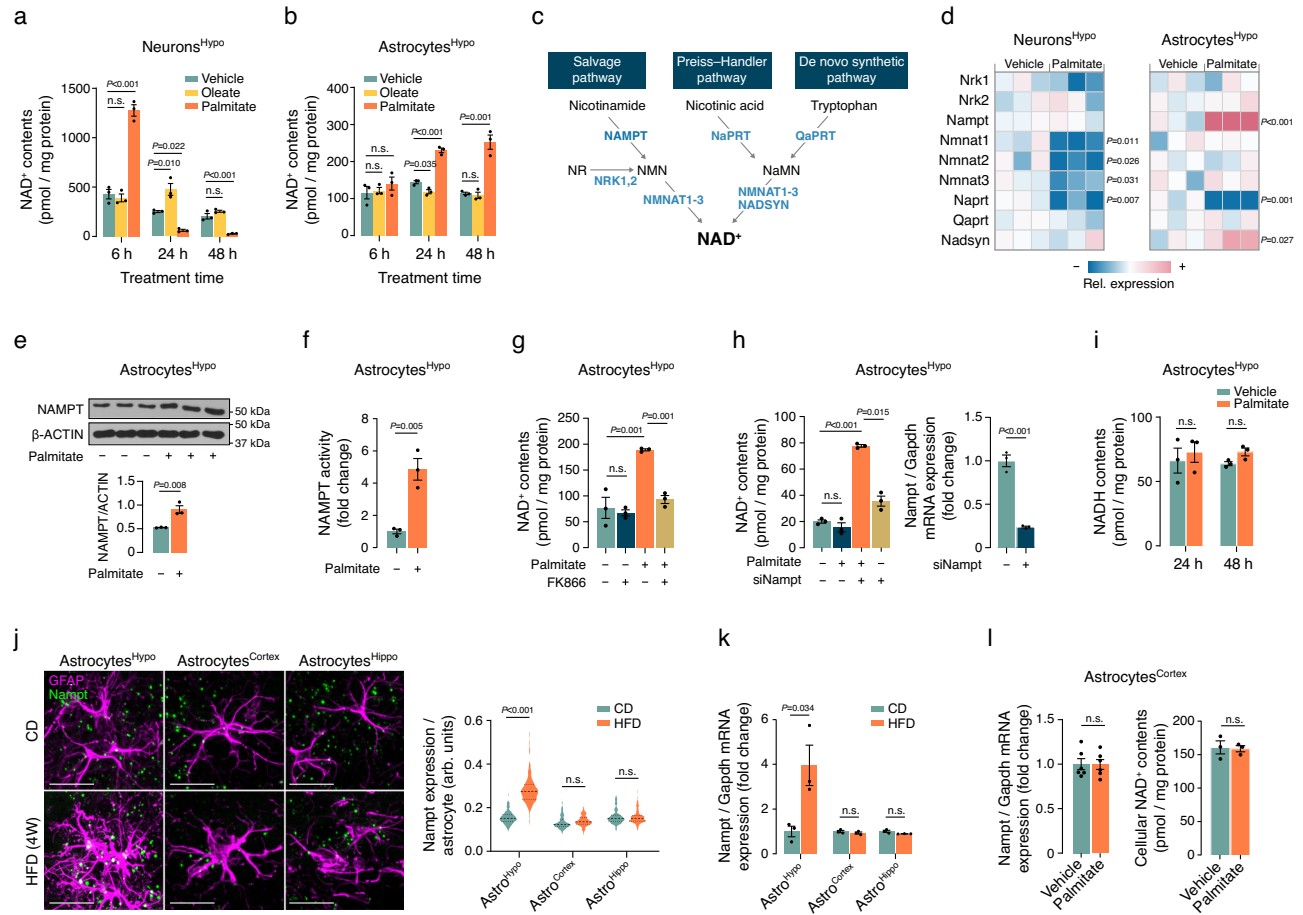

**Fig. 1 | Saturated fat overload activates the NAD+ salvage pathway in hypothalamic astrocytes. a, b** Effects of oleate or palmitate treatment (200 μM each for the indicated duration) on cellular NAD+ contents in N1 hypothalamic neuronal cells or primary cultured hypothalamic astrocytes (*n* = 3 wells). **c** Schematic illustration depicting the various NAD+ biosynthetic pathways. **d** Heatmaps showing the palmitate treatment (200 μM, 48 h)-induced alterations in expression levels of NAD+ biosynthetic enzymes in hypothalamic neurons and astrocytes (*n* = 3 wells). **e, f** Changes in nicotinamide phosphoribosyltransferase (NAMPT) protein expression and enzyme activity in hypothalamic astrocytes following palmitate treatment (200 μM, 48 h) (*n* = 3 wells). **g, h** Effects of NAMPT inhibition using FK866 (500 nM, 48 h) or Nampt siRNA treatment on the palmitate-induced increase in the cellular NAD+ levels of hypothalamic astrocytes (*n* = 3 wells). **i** Palmitate treatment (200 μM) did not alter cellular NADH levels in primary hypothalamic astrocytes

(*n* = 3 wells). **j** Nampt fluorescence in situ hybridization (FISH) and GFAP dual staining showing the changes in Nampt mRNA expression in hypothalamic, cortical, and hippocampal astrocytes of 15-week-old C57 male mice fed either a CD or HFD for 4 weeks (*n* = 120 astrocytes from 4 mice per group). Scale bars: 20 μm. arb. units: arbitrary unit. **k** qPCR analysis of Nampt expression in astrocytes isolated from different brain regions (hypothalamus, cortex, and hippocampus) of 15-week-old C57 male mice subjected to a CD or HFD for 4 weeks (*n* = 3). **l** Palmitate treatment did not induce alterations in Nampt mRNA expression or cellular NAD+ content in primary cortical astrocytes (Nampt, *n* = 6; NAD+, *n* = 3). One-way ANOVA followed by Fisher's LSD test (**a, b, g, h**-NAD+) and two-sided unpaired t-test (**d–f, h**- Nampt, **i–l**). n.s.: not significant. Three independent replicates were performed for all studies and measurement were taken from distinct samples. Results are presented as the mean ± SEM. Source data are provided as a Source Data file.

palmitate treatment failed to increase cellular NAD+ content and Nampt expression in primary cultured cortical astrocytes (Fig. 1l). These results suggest that activation of the NAD+ salvage pathway due to excessive fat intake specifically occurs in hypothalamic astrocytes.

### Inhibition of the astrocyte NAD+ salvage pathway mitigates HFD-induced obesity

Given the activation of the hypothalamic astrocytic NAD+ salvage pathway due to excessive fat intake, we explored the role of this pathway in the development of DIO. To specifically inhibit the NAD+ salvage pathway in astrocytes, we generated mice with astrocyte-specific NAMPT depletion, referred to as Nampt-AKO or AKO mice. Previous studies have demonstrated astrocyte-specific gene deletion in the mediobasal hypothalamus using tamoxifen-inducible human GFAP-Cre (hGFAP-Cre[ERT2]) mice[10,32]. Thus, AKO mice were created by crossbreeding hGFAP-Cre[ERT2] mice with Nampt-floxed (Nampt[f/f]) mice and inducing gene knockout through tamoxifen injections (100 mg/kg/day for 5 days) at 7 weeks of age (Fig. 2a). Successful Nampt

knockdown in hypothalamic astrocytes was confirmed through Nampt FISH and GFAP double staining and qPCR analysis of astrocytes isolated from the mediobasal hypothalamus (MBH) (Fig. 2b, c). Similarly, the NAD+ content in isolated MBH astrocytes was significantly lower in AKO mice than in WT mice (Fig. 2c). Because neither hGFAP-Cre[ERT2] nor Nampt[f/f] mice injected with tamoxifen showed no significant differences in metabolic phenotypes under both CD- and HFD-fed conditions (Supplementary Fig. 2a, b), we used these mice as a wild-type (WT) control.

Body weight monitoring showed no significant difference in body weights between WT and AKO male mice during the CD phase (Fig. 2d). However, during the 12-week HFD challenge, AKO mice exhibited 8%–10% lower body weights compared to their sex-matched WT littermates (Fig. 2d). Body composition analysis revealed a 22% reduction in fat mass in AKO mice fed a HFD for 4 weeks, with no discernible changes in lean mass (Fig. 2e). This finding was confirmed by magnetic resonance imaging (MRI), which illustrated diminished subcutaneous and visceral fat mass in AKO males (Fig. 2f).

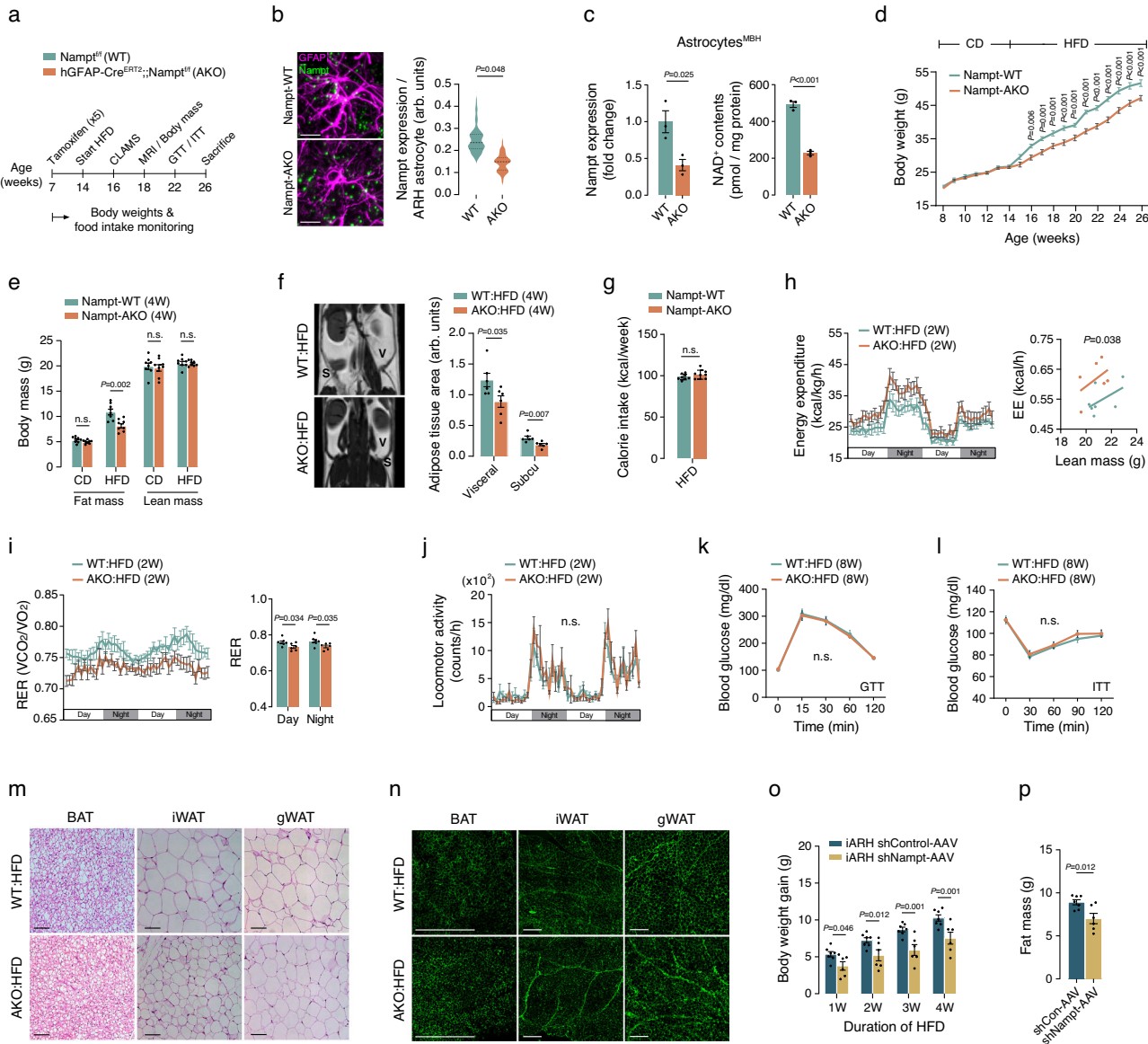

**Fig. 2 | Blockade of the astrocyte NAD⁺ salvage pathway attenuates HFD-induced obesity. a** Schematic illustrating metabolic phenotyping studies in male mice with astrocyte-specific NAMPT depletion. **b** Nampt FISH and GFAP double staining showing successful Nampt deletion in GFAP⁺ ARH astrocytes in 16-week-old Nampt-AKO male mice compared to that in age- and sex-matched Nampt-WT mice ($n = 90$ astrocytes from 3 mice per group). Scale bars: 20 µm. arb. units: arbitrary unit. **c** qPCR analysis of Nampt expression and measurement of NAD⁺ contents in astrocytes isolated from the mediobasal hypothalamus (MBH) of 16-week-old Nampt-WT or Nampt-AKO mice ($n = 3$ mice). **d** Comparison of body weights after tamoxifen injection between Nampt-AKO and Nampt-WT male mice during sequential feeding with a CD and HFD ($n = 8$ mice). **e, f** Body mass analysis of 18-week-old Nampt-WT and Nampt-AKO male mice using two methods: dual energy absorptiometry (**e**, $n = 8$) and MRI (**f**, $n = 6$). S: subcutaneous fat, V: visceral fat. arb. units: arbitrary unit. **g** Calorie intakes during HFD feeding in male Nampt-AKO and Nampt-WT mice ($n = 8$ mice). **h–j** Energy expenditure (EE), respiratory exchange ratio (RER), and locomotor activity in 16-week-old Nampt-AKO and Nampt-WT mice

(EE and RER, AKO, $n = 6$ mice, WT, $n = 7$ mice; Locomotor activity, AKO, $n = 5$ mice, WT, $n = 6$ mice). **k, l** Glucose and insulin tolerance tests (GTT, ITT) in 22-week-old male Nampt-AKO and Nampt-WT mice ($n = 8$ mice). **m** H & E staining performed on brown adipose tissue (BAT), inguinal white adipose tissue (iWAT), and gonadal white adipose tissues (gWAT) from 18-week-old Nampt-AKO and Nampt-WT mice ($n = 5$ mice). Scale bars: 100 µm. **n** Comparison of sympathetic innervation in adipose tissues between 18-week-old Nampt-AKO male mice and their WT littermates ($n = 5$ mice). Scale bars: 500 µm. **o, p** Effects of ARH astrocyte-specific Nampt knockdown through intra-ARH (iARH) injection of Nampt shRNA (shNampt)-AAV on HFD-induced weight gain and fat mass in 16-week-old hGFAP-Cre^ERT2 male mice (shNampt, $n = 6$ mice; shControl, $n = 7$ mice). Two-sided unpaired t-test (**b, c, e–g, i, p**) and two-way repeated measures ANOVA followed by Fisher's LSD test (**d, j–l, o**) and one-way ANCOVA using lean mass as covariate (**h**). n.s.: not significant. Three independent replicates were performed for all studies and measurement were taken from distinct samples. Results are presented as the mean ± SEM. Source data are provided as a Source Data file.

To further elucidate the mechanisms responsible for obesity resistance in AKO mice, we investigated their metabolic changes during HFD feeding. Despite an unchanged calorie intake, AKO males displayed elevated energy expenditure (EE) (Fig. 2g, h). In addition, the respiratory exchange ratio (RER), which reflects fuel source utilization, was lower in AKO mice, indicating a preference for fat over carbohydrate as a fuel source (Fig. 2i). In contrast, there was no significant

difference in locomotor activity, glucose tolerance, or insulin sensitivity between male AKO and WT mice on a HFD (Fig. 2j–l). Moreover, under CD-fed conditions, AKO males displayed no alterations in food intake, EE, RER, locomotor activity, or glucose and insulin tolerance (Supplementary Fig. 2c–h). CD- or HFD-fed AKO females showed no significant changes in the aforementioned metabolic parameters (Supplementary Fig. 3).

Histological examination of adipose depots in AKO mice subjected to a HFD revealed a significant decrease in lipid droplet sizes in brown adipose tissue (BAT) and adipocyte sizes across all adipose depots, although the browning phenomenon was absent in white adipose tissues (Fig. 2m and Supplementary Fig. 2g). Considering the role of hypothalamic astrocytes in regulating adipose tissue lipid metabolism via sympathetic neural control[33], we next assessed adipose tissue sympathetic innervation using tyrosine hydroxylase (TH) staining. Compared to WT mice, AKO mice exhibited significantly higher TH+ sympathetic nerve fiber densities in multiple fat depots (Fig. 2n and Supplementary Fig. 2h). Therefore, under conditions of overnutrition, activation of the astrocyte NAD+ salvage pathway may enhance adipose tissue fat deposition by reducing sympathetic activity.

We conducted a further investigation to determine whether deleting Nampt specifically in ARH astrocytes could induce metabolic changes similar to those observed in AKO mice. To achieve this, we generated adeno-associated virus (AAV) vectors that expressed both Nampt shRNA and GFP (shNampt-AAV) in a Cre-dependent manner. Subsequently, we bilaterally administered these vectors into the ARH of hGFAP-Cre[ERT2] male mice 2 weeks after tamoxifen injections. As the control group, male hGFAP-Cre[ERT2] mice received an equal amount of AAV carrying non-targeting shRNA and GFP (shControl-AAV). We confirmed the efficacy of AAV infection by observing GFP expression confined to the astrocytes in the ARH, which showed no significant difference between the shNampt-AAV and shControl-AAV groups (Supplementary Fig. 4a, b). Nampt knockdown in ARH astrocytes was confirmed by the reduced Nampt expression in infected hypothalamic astrocytes, whereas unaltered Nampt expression was observed in non-infected ARH astrocytes (Supplementary Fig. 4c). Beginning 2 weeks after AAV injections, the mice were subjected to a HFD challenge, and their body weight was monitored over the 4-week HFD-feeding period. Similar to what was observed in AKO mice, the mice that received shNampt-AAV exhibited reduced weight gain and lower fat mass compared to those that received shControl-AAV (Fig. 2o, p). These findings underscore the significant contribution of the ARH astrocyte NAD+ salvage pathway to the development of DIO.

## NAD+ salvage pathway activates proinflammatory programs in hypothalamic astrocytes

Hypothalamic inflammation, characterized by activated microgliosis, reactive astrogliosis, and enhanced expression of proinflammatory cytokines, has been proposed as a fundamental mechanism underlying obesity-related hypothalamic dysfunction[1]. Therefore, we compared HFD-induced hypothalamic inflammation between Nampt-AKO and -WT male mice exposed to a HFD for 4 weeks. Remarkably, HFD-triggered activation of microglia was significantly reduced in the ARH of AKO mice compared to their WT counterparts, as evidenced by the reduced number of IBA1+ activated microglia and the simultaneous increase in the number of TMEM119+ quiescent microglia (Fig. 3a). Reactive astrogliosis was also assessed using GFAP and complement C3 double staining, revealing a substantial decrease in the accumulation of GFAP+ C3high reactive astrocytes in the ARH of AKO mice on a HFD (Fig. 3b). Furthermore, AKO mice exhibited a reduction in the MBH mRNA expression of proinflammatory genes such as proinflammatory cytokines (Il-1β, Il-6, Tnf-α), chemokines (Mcp-1, Mip-2), and proinflammatory cell markers (Gfap, C3, S100β, Iba1) (Fig. 3c). These results suggest that inhibition of the astrocytic NAD+ salvage pathway suppresses the hypothalamic innate immune/inflammatory responses to dietary fats.

Because hypothalamic inflammation impairs leptin signaling[34], we next compared leptin signaling in ARH neurons between 2-week HFD-fed AKO and WT mice. The comparison involved the intraperitoneal injection of leptin, followed by the measurement of phospho-STAT3 expression in the ARH at 30 min after injection. To minimize the

influence of adiposity on hypothalamic neuronal responses to leptin, we conducted a leptin response study in WT and AKO mice with comparable body weights and fat mass (Supplementary Fig. 5a). The phosphorylation of STAT3 induced by leptin in ARH neurons was more prominent in AKO mice than in WT controls, suggesting an enhanced leptin response in the hypothalamus of AKO mice (Fig. 3d). To address the improvement in leptin sensitivity observed in AKO mice, we examined changes in the molecules involved in leptin signal transduction, such as the functional leptin receptor Leprb and the leptin signaling inhibitors Ptp1b and Socs3 in AKO mice. We observed a significant reduction in the expression of Ptp1b and Socs3, while Leprb expression was unaltered (Fig. 3e). Given that hypothalamic Ptp1b and Socs3 expression increase under HFD-induced inflammatory conditions[35,36], the reduced hypothalamic inflammation may enhance leptin signaling by lowering Ptp1b and Socs3 expression.

Hypothalamic astrocytes regulate adipose tissue lipolysis through proopiomelanocortin (POMC) neurons[33]. Thus, we evaluated the activity of POMC neurons under freely-fed conditions in AKO and WT mice, which were fed a HFD for 4 weeks, using Pomc FISH and c-Fos double staining. The results revealed an increase in the percentages of c-Fos+ POMC neurons in AKO mice compared to WT mice (Fig. 3f). Taken together, these results suggest that the HFD-induced activation of the NAD+ salvage pathway in astrocytes contributes to HFD-induced hypothalamic inflammation, resulting in leptin resistance and reduction in POMC neuronal activity.

To substantiate the direct involvement of the NAD+ salvage pathway in hypothalamic inflammation, we artificially activated the NAD+ salvage pathway by overexpressing the Nampt gene in cultured mouse hypothalamic astrocytes and inhibited it with FK866 cotreatment. The overexpression (O/E) of Nampt indeed resulted in increased NAMPT activity and elevated cellular NAD+ levels in hypothalamic astrocytes, both of which were reversed by cotreatment with FK866 (Supplementary Fig. 5b). We then assessed the expression of proinflammatory genes. We found that Nampt O/E led to the upregulation of proinflammatory cytokines (Il-1β, Il-6, Tnf-α) and the chemokine Mcp-1, along with increased secretion of TNF-α by hypothalamic astrocytes (Fig. 3g, h). Importantly, the alterations triggered by Nampt O/E were effectively suppressed by the administration of FK866 (Fig. 3g, h), confirming that these effects were directly attributable to the upregulation of NAMPT enzyme activity. These results provide compelling evidence of the proinflammatory role of the NAD+ salvage pathway in hypothalamic astrocytes. However, NAMPT inhibition with FK866 only partially reversed the effect of palmitate treatment on Il-6 and Mcp-1 expression (Fig. 3g, h). These data suggest that palmitate upregulates Il-6 and Mcp-1 expression via both NAMPT-dependent and -independent pathways, such as TLR4−NF-κB signaling and inflammasome activation[37,38].

Interactions between neurons and glia, as well as among different types of glia cells, are believed to play a crucial role in hypothalamic inflammatory responses to dietary fats[39,40]. Thus, we hypothesized that astrocytes with an activated NAD+ salvage pathway facilitate the spread of inflammation to neighboring microglia and neurons by releasing proinflammatory molecules. To investigate this, we exposed hypothalamic neurons and microglia to a conditioned medium derived from hypothalamic astrocytes (referred to as ACM), with or without Nampt O/E. Treatment with Nampt O/E-ACM increased the expression of proinflammatory genes in primary cultured hypothalamic microglia and neurons (Fig. 3i). Notably, NAMPT can be secreted into body fluids, and this extracellular NAMPT (eNAMPT) exhibits characteristics similar to proinflammatory cytokines[19,21]. To determine whether eNAMPT released by astrocytes is responsible for inducing hypothalamic inflammation, we next attempted to counteract the effects of eNAMPT by administering a NAMPT neutralizing antibody alongside Nampt O/E-ACM. This intervention attenuated the production of proinflammatory cytokines induced by Nampt O/E-ACM in both

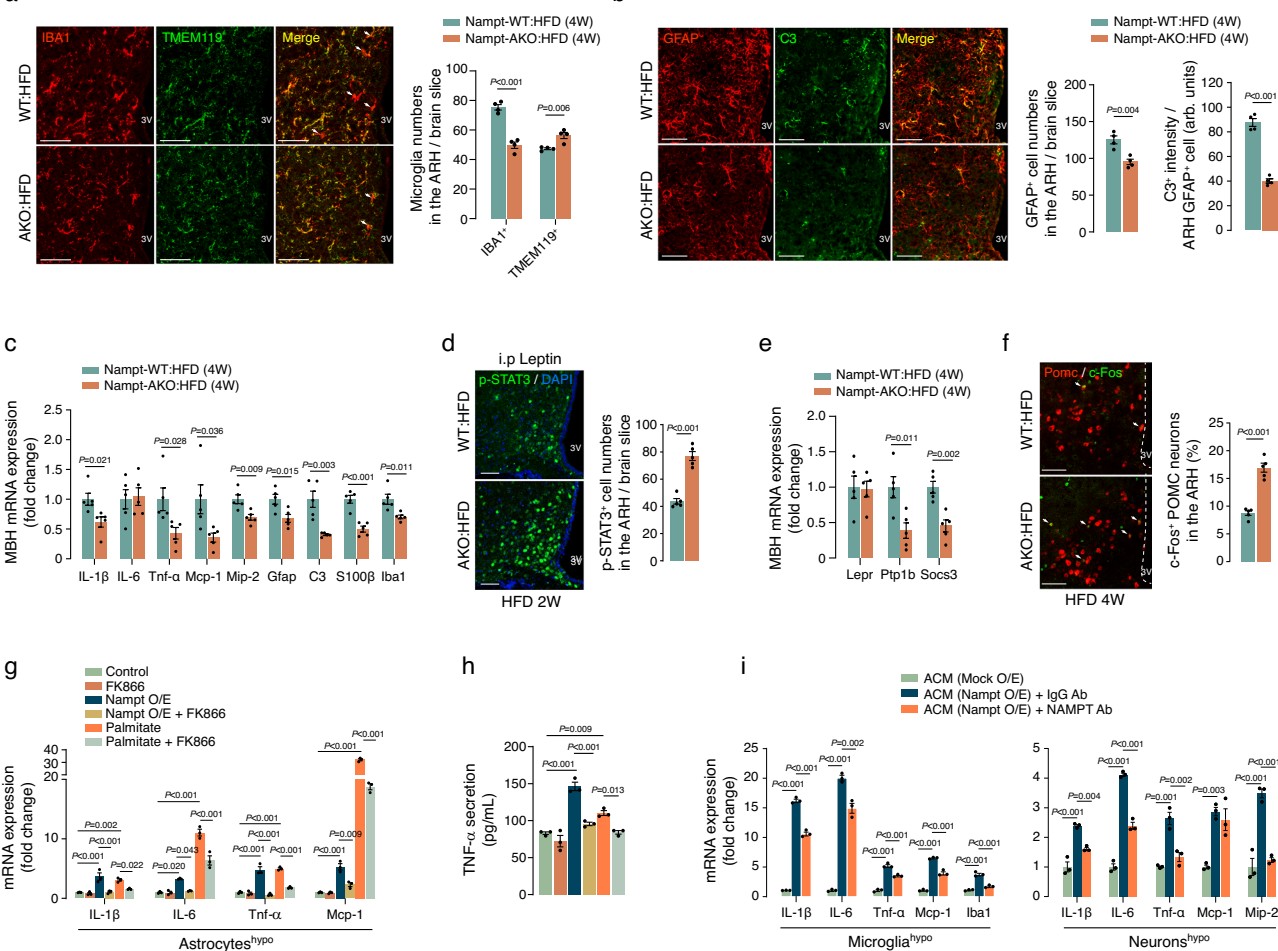

**Fig. 3 | NAD⁺ salvage pathway triggers proinflammatory programs in hypothalamic astrocytes. a** Phenotypic change in the ARH microglia of 18-week-old male Nampt-AKO mice on a HFD compared to those in age- and sex-matched Nampt-WT littermates. IBA1 and TMEM119 immunostaining was used to identify activated and quiescent microglia, respectively (*n* = 4 mice). Scale bars: 50 µm. 3 V: Third ventricle. **b** Double immunostaining of GFAP (astrocyte marker) and C3 (reactive astrocyte marker) showing the activation state of ARH astrocytes in 18-week-old male Nampt-AKO and Nampt-WT mice after 4 weeks on a HFD (*n* = 4 mice). Scale bars: 50 µm. **c** qPCR analysis to assess the expression of proinflammatory markers in the MBH of 18-week-old Nampt-AKO and Nampt-WT mice fed a HFD for 4 weeks (*n* = 5 mice). **d** Comparison of leptin-induced STAT3 phosphorylation in the ARH of 16-week-old Nampt-AKO and Nampt-WT males after 2 weeks on a HFD (*n* = 5 mice). Scale bars: 50 µm. **e** qPCR analysis of leptin receptor and leptin signaling regulators in the hypothalamus of 18-week-old Nampt-AKO and

Nampt-WT males on an HFD for 4 weeks (*n* = 5 mice). **f** Assessment of POMC neuronal activity in 18-week-old Nampt-AKO and Nampt-WT male mice after 4 weeks on a HFD, under freely fed conditions, via Pomc FISH and c-Fos double staining (*n* = 5 mice). Scale bars: 50 µm. **g**, **h** Effects of Nampt overexpression (O/E) and palmitate treatment (200 µM), and their combination with FK866 (500 nM) on proinflammatory cytokine/chemokine expression and TNF-α secretion from primary hypothalamic astrocytes (*n* = 3 wells). **i** Effects of astrocyte-conditioned medium (ACM) treatment with or without a 1 µg NAMPT neutralizing antibody on the expression of proinflammatory markers in primary hypothalamic neurons and microglia (*n* = 3 wells). ACM was collected from hypothalamic astrocytes with or without Nampt O/E. Two-sided unpaired t-test (**a**–**f**) and one-way ANOVA followed by Fisher's LSD test (**g**–**i**). Two independent replicates were performed for all studies and measurement were taken from distinct samples. Results are presented as the mean ± SEM. Source data are provided as a Source Data file.

hypothalamic microglia and neurons (Fig. 3i). Therefore, it is plausible that eNAMPT released by reactive astrocytes could induce hypothalamic inflammation through a paracrine mechanism. However, because NAMPT antibody treatment partially inhibited the effect (Fig. 3i), we conducted further investigations into the blocking efficacy of NAMPT neutralizing antibody on the effects of Nampt O/E-ACM by measuring cellular NAD⁺ levels in microglia and neurons. Our results demonstrated that treatment with Nampt O/E-ACM increased NAD⁺ levels in both hypothalamic neurons and microglia (Supplementary Fig. 5c). This increase was blocked by cotreatment with the NAMPT neutralizing antibody, demonstrating the blocking efficacy of the NAMPT antibody (Supplementary Fig. 5c). These findings suggest the possibility that inflammatory factors other than eNAMPT may also mediate Nampt O/E-ACM-induced inflammation in neurons and microglia.

## CD38 acts as a downstream mediator of the NAD⁺ salvage pathway in inflammatory astrocytes

We next conducted further experiments to identify a downstream mediator of the NAD⁺ salvage pathway in inflammatory astrocytes. Elevated NAD⁺ levels resulting from the activation of the NAD⁺ salvage pathway can trigger the activation of NAD⁺-dependent enzymes, such as SIRTs, PARP, and CD38[16]. Therefore, we investigated whether inhibition of these enzymes could mitigate cytokine and chemokine production induced by Nampt O/E in hypothalamic astrocytes. Treatment with the SIRT1/2 chemical inhibitor sirtinol or gene knockdown of Sirt-1, 2, 6 failed to impede Nampt O/E-triggered increases in Il-1β, Tnf-α, and Mcp-1 expression in primary hypothalamic cultured astrocytes (Supplementary Fig. 6a–d). Similarly, administration of the PARP inhibitor olaparib was unable to reverse the proinflammatory effects of Nampt O/E (Supplementary Fig. 6e). In contrast, the CD38 inhibitor 78c

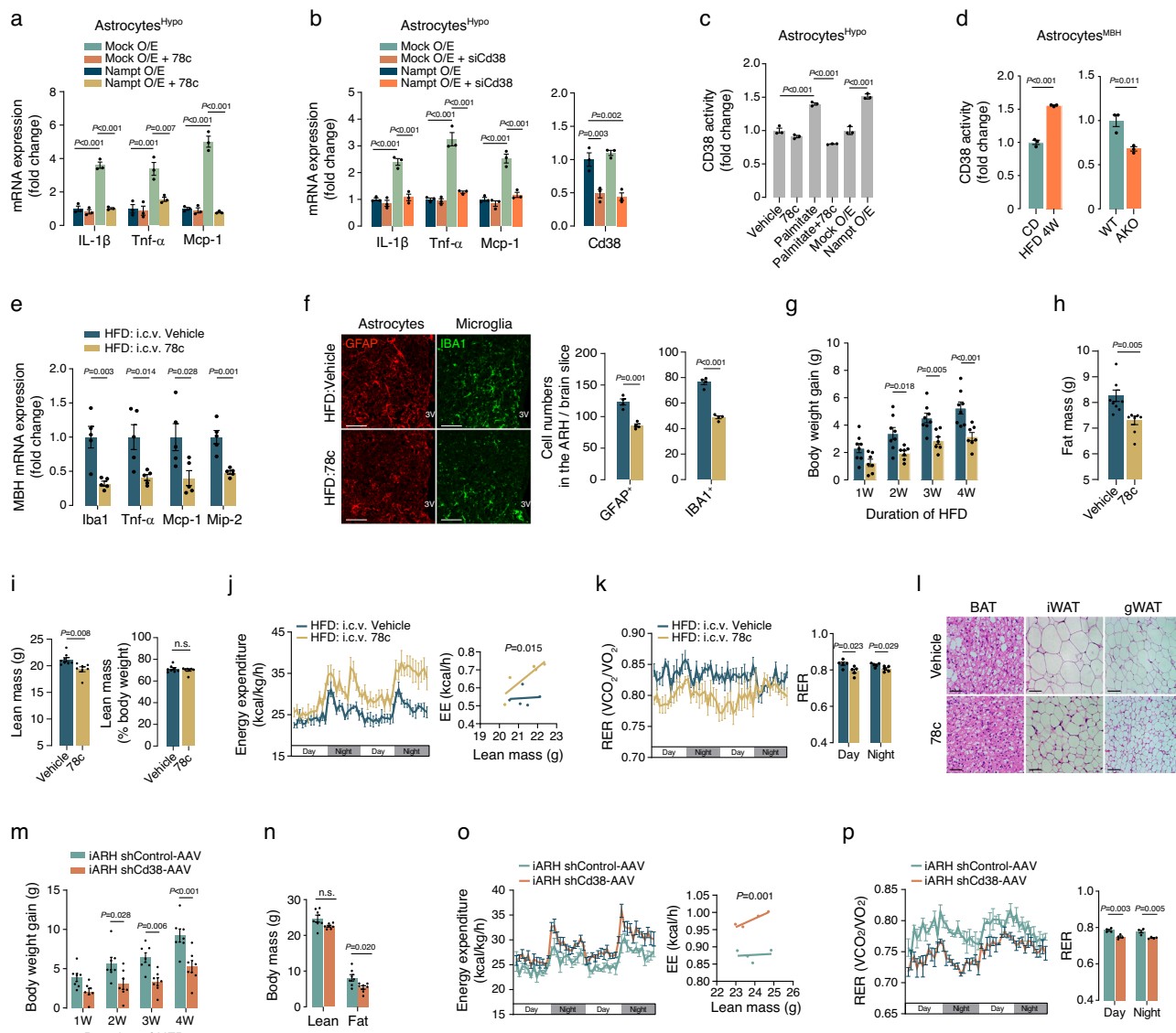

**Fig. 4 | CD38 acts as a downstream mediator of astrocyte inflammation induced by the NAD+ salvage pathway. a, b** Effects of CD38 blockade via 78c treatment (100 nM, 48 h) (**a**) and Cd38 siRNA (siCd38) (**b**) on Nampt O/E-induced proinflammatory cytokine/chemokine expression in primary hypothalamic astrocytes derived from C57 male mice ($n = 3$ mice). **c** Effects of palmitate treatment with or without cotreatment of 78c and Nampt O/E on CD38 activity in primary hypothalamic astrocytes ($n = 3$ wells). **d** CD38 enzyme activity in hypothalamic astrocytes isolated from the MBH of 16-week-old C57 male mice fed a CD or HFD for 4 weeks or from the MBH of 16-week-old Nampt-AKO and Nampt-WT males on a 4-week HFD ($n = 3$ mice). **e** Effect of 4 week-intracerebroventricular (i.c.v.) infusion of 78c on the hypothalamic inflammatory gene expression in 16-week-old C57 male mice during HFD feeding ($n = 5$). **f** Effects of 78c-induced hypothalamic CD38 inhibition on reactive astrogliosis and activated microgliosis in the ARH of 16-week-old C57 male mice fed an HFD for 4 weeks ($n = 4$ mice). Scale bars: 50 μm. **g–i** Comparison of weight gain, fat mass, and lean mass between 16-week-old C57 male mice that received i.c.v. infusion of 78c and those that received i.c.v. vehicle infusion (vehicle, $n = 8$ mice; 78c, $n = 7$ mice). **j, k** Effects of hypothalamic CD38 inhibition with 78c infusion on EE and RER in HFD-fed, 14-week-old male mice ($n = 5$). **l** Adipose tissue histology in the BAT, iWAT, and gWAT of mice subjected to i.c.v. infusion of either vehicle or 78c in 16-week-old C57 male mice. Scale bars: 50 μm. **m–p** Effects of ARH astrocyte-specific CD38 knockdown on the body weight gain, body mass, EE, and RER in 14 ~ 16 weeks-old male hGFAP-Cre[ERT2] mice that received bilateral iARH injection of Cd38 shRNA (shCd38)-AAV and were subjected to a HFD (**m**, **n**, $n = 7$ mice; **o**, **p**, $n = 4$ mice). Two-sided unpaired t-test (**d–f**, **h**, **i**, **k**, **n**, **p**), one-way ANOVA (**a–c**) and two-way repeated measures ANOVA (**g**, **m**) followed by Fisher's LSD test, and one-way ANCOVA using lean mass as covariate (**j**, **o**). n.s.: not significant. Two independent replicates were performed for all studies and measurement were taken from distinct samples. Results are presented as the mean ± SEM. Source data are provided as a Source Data file.

and Cd38 knockdown effectively blocked the effects of Nampt O/E on inflammatory cytokine production, although treatment with either of them alone did not reduce the basal expression of inflammatory cytokines (Fig. 4a, b). These findings strongly suggest that CD38 operates downstream of the NAD+ salvage pathway in the context of astrocyte inflammation induced by fat accumulation. To reinforce this hypothesis, palmitate treatment, and Nampt O/E increased CD38 activity in cultured hypothalamic astrocytes (Fig. 4c). Moreover, cotreatment with 78c attenuated the palmitate-induced increases in

CD38 activity, although treatment with 78c alone had no significant effect on basal CD38 activity (Fig. 4c). In addition, we found increased CD38 activity in MBH astrocytes isolated from male mice on a HFD for 4 weeks, whereas it was reduced in HFD-fed AKO male mice compared with their WT counterparts (Fig. 4d).

Next, we examined the effects of hypothalamic CD38 activity inhibition on hypothalamic inflammation and obesity induced by excessive fat intake. To accomplish this, C57BL/6J male mice were subjected to a 4-week infusion of either 78c or vehicle through a

cerebroventricle-implanted cannula connected to subcutaneous osmotic pumps during HFD feeding. We confirmed a significant reduction in hypothalamic CD38 activity in mice treated with 78c (Supplementary Fig. 7a). In this group, we observed a substantial reduction in the expression levels of Iba1, Tnf-α, Mcp-1, and Mip-2 in the MBH, along with a significant reduction in reactive astrogliosis and activated microgliosis (Fig. 4e, f). Moreover, the administration of 78c suppressed HFD-induced weight gain and reduced fat mass although HFD-induced weight gain was lower in this study due to osmotic pump insertion and intracerebroventricular (i.c.v) cannulation (Fig. 4g, h). In mice treated with i.c.v. 78c, lean mass, measured in grams, was reduced; however, when expressed as a percentage of body weight, it remained unaltered (Fig. 4i). These effects were accompanied by increased EE and decreased RER (Fig. 4j, k). However, the food intake and locomotor activity of these mice remained unchanged (Supplementary Fig. 7b, c). Histological analysis of adipose tissue revealed diminished fat droplet sizes and increased sympathetic innervation in BAT, inguinal white adipose tissue (iWAT), and gonadal white adipose tissue (gWAT) of mice treated with 78c compared to those injected with the vehicle (Fig. 4l and Supplementary 7d, e). Remarkably, all of the observed outcomes of 78c treatment resembled the phenotype exhibited by Nampt-AKO mice (Fig. 2).

To specifically inhibit CD38 activity in ARH astrocytes, we administered AAV expressing Cd38 shRNA (shCd38-AAV) in a Cre-dependent manner to both sides of the ARH in male hGFAP-Cre$^{ERT2}$ mice, while control mice received the same dosage of AAV-expressing control shRNA (shControl-AAV). The effectiveness of CD38 knockout and AAV infections was established in ARH astrocytes (Supplementary Fig. 8). The specific expression of shCD38 in ARH astrocytes diminished weight gain, reduced fat mass, increased EE, and lowered RER during HFD consumption (Fig. 4m–p). Taken together, these findings suggest that CD38 activation in ARH astrocytes contributes to DIO by impeding peripheral energy expenditure and fat utilization.

### Fat overload disrupts calcium signals in hypothalamic astrocytes via the NAMPT-NAD$^+$-CD38 axis

Astrocyte calcium (Ca$^2$) transients play a critical role in the regulation of gliotransmitter release and cerebral blood flow[41]. Intracellular calcium concentrations (i[Ca$^{2+}$]) in astrocytes exhibit spontaneous oscillations and respond to various physiological and pathophysiological stimuli[41]. Importantly, CD38 is responsible for converting NAD$^+$ into the Ca$^{2+}$-mobilizing metabolites cyclic adenosine diphosphate (ADP)-ribose (cADPR) and ADP-ribose (ADPR)[42,43]. Therefore, it is plausible that the activation of the NAD$^+$ salvage–CD38 axis by fat overload could influence i[Ca$^{2+}$] dynamics in hypothalamic astrocytes through the release of cADPR and ADPR. To test this hypothesis, we determined the cellular levels of cADPR and ADPR in hypothalamic astrocytes treated with palmitate or overexpressing Nampt and Cd38. We found that these treatments significantly increased the cellular cADPR and ADPR contents in cultured hypothalamic astrocytes (Fig. 5a, b).

We further explored the dynamics of i[Ca$^{2+}$] in hypothalamic astrocytes using the fluorescent resonance energy transfer (FRET)-based Ca$^{2+}$ sensor YC3.60. The results revealed that treatment with palmitate robustly stimulated i[Ca$^{2+}$] oscillations, which were suppressed by administration of 78c or by knockdown of Nampt or Cd38 (Fig. 5c, d). In line with this, the basal i[Ca$^{2+}$] levels were significantly elevated in hypothalamic astrocytes overexpressing Nampt and Cd38 (Fig. 5e). Collectively, these findings demonstrate that Ca$^{2+}$ signals in hypothalamic astrocytes are markedly activated in response to fat overload and that the NAMPT–NAD$^+$–CD38 axis plays a pivotal role in driving this response.

Given that astrocyte foot processes extensively envelop blood vessels[44], they likely play a vital role in sensing circulating blood hormone signals and transmitting this information to hypothalamic neurons. Notably, key metabolic hormones, such as leptin, insulin, and glucagon-like peptide (GLP-1), exert their actions on hypothalamic astrocytes, thereby influencing the hypothalamic regulation of energy and glucose metabolism[10,32,45]. Therefore, we next conducted experiments to determine whether these metabolic hormones could induce Ca$^{2+}$ signals in hypothalamic astrocytes and whether these Ca$^{2+}$ responses were altered in fat-overloaded astrocytes. Our results revealed that treatment with leptin, insulin, and GLP-1 promptly elicited Ca$^{2+}$ signals in hypothalamic astrocytes (Fig. 5f–h). Importantly, these responses were diminished in hypothalamic astrocytes pre-exposed to palmitate for 48 h (Fig. 5f–h). These findings suggest that chronic consumption of fat-rich diets impairs the appropriate Ca$^{2+}$ response to metabolic hormones in hypothalamic astrocytes. Because astrocyte Ca$^{2+}$ signals critically affect neuronal and synaptic functions by controlling gliotransmitter release, this impairment in hormonal Ca$^{2+}$ signaling in hypothalamic astrocytes may ultimately disrupt energy homeostasis and contribute to the development of obesity.

## Discussion

Our study reveals divergent effects of saturated fat overload on cellular NAD$^+$ levels and the expression of NAD$^+$ biosynthetic enzymes in hypothalamic neurons and astrocytes. In hypothalamic astrocytes, saturated fat overload increases NAD$^+$ levels by upregulating Nampt expression, while suppressing NAD$^+$ biosynthesis in hypothalamic neurons by downregulating Nmnat expression. Interestingly, short-term exposure (6 h) to palmitate increased the cellular NAD$^+$ content in hypothalamic neurons, as reported previously[46]. Hence, the NAD$^+$ biosynthetic pathway in hypothalamic neurons may exhibit dual responses to saturated fatty acids, depending on the duration of exposure.

Accumulating evidence strongly suggests that astrocytes display significant heterogeneity across various brain regions in terms of morphology and function[47]. Our study showed brain region-specific diversity among astrocytes regarding their response to fatty acids in modulating the NAD$^+$ salvage pathway. Upon exposure to palmitate, Nampt expression was elevated in hypothalamic astrocytes; however, no such effect was observed in astrocytes from the cortical and hippocampal regions. These differing responses of astrocytes may underlie the increased vulnerability of the hypothalamus to neuroinflammatory damage induced by excessive fat intake. It is also quite possible that astrocytes within different nuclei of the hypothalamus exhibit functional variability and distinct responses to palmitate or a HFD. However, our findings, based on MBH measurements, could not address this possibility.

In our study, the male mice with depleted Nampt in astrocytes exhibited no significant change in metabolic phenotype when fed a CD. However, when exposed to a HFD, they displayed resistance to weight gain and adipose tissue fat accumulation. Considering that current treatments for human obesity typically target more than 5% weight loss, the effect of weight loss due to astrocyte Nampt depletion may not be substantial but still physiologically accountable. This obesity resistance is attributed to increased EE and reduced RER, implying enhanced fat oxidation. A recent study has shown that chemogenetic stimulation of ARH astrocytes enhances adipose tissue lipolysis through increased sympathetic outflow to adipose tissue[33]. Nampt-AKO mice consistently showed increased sympathetic innervation in various fat depots. These findings suggest that under conditions of fat overload, activation of the astrocytic NAD$^+$ salvage pathway may restrain the EE and fat oxidation in adipose tissue by downregulating sympathetic nerve innervation, which may eventually lead to the accumulation of adipose tissue fat and the development of obesity. It has also been shown that the modulation of hypothalamic astrocyte activity influences food intake[48,49] and glucose homeostasis[50]. However, in our study, chronic inhibition of the astrocyte NAD$^+$ salvage pathway did not significantly alter food intake, glucose tolerance, or insulin sensitivity. The preferential use of fat as a

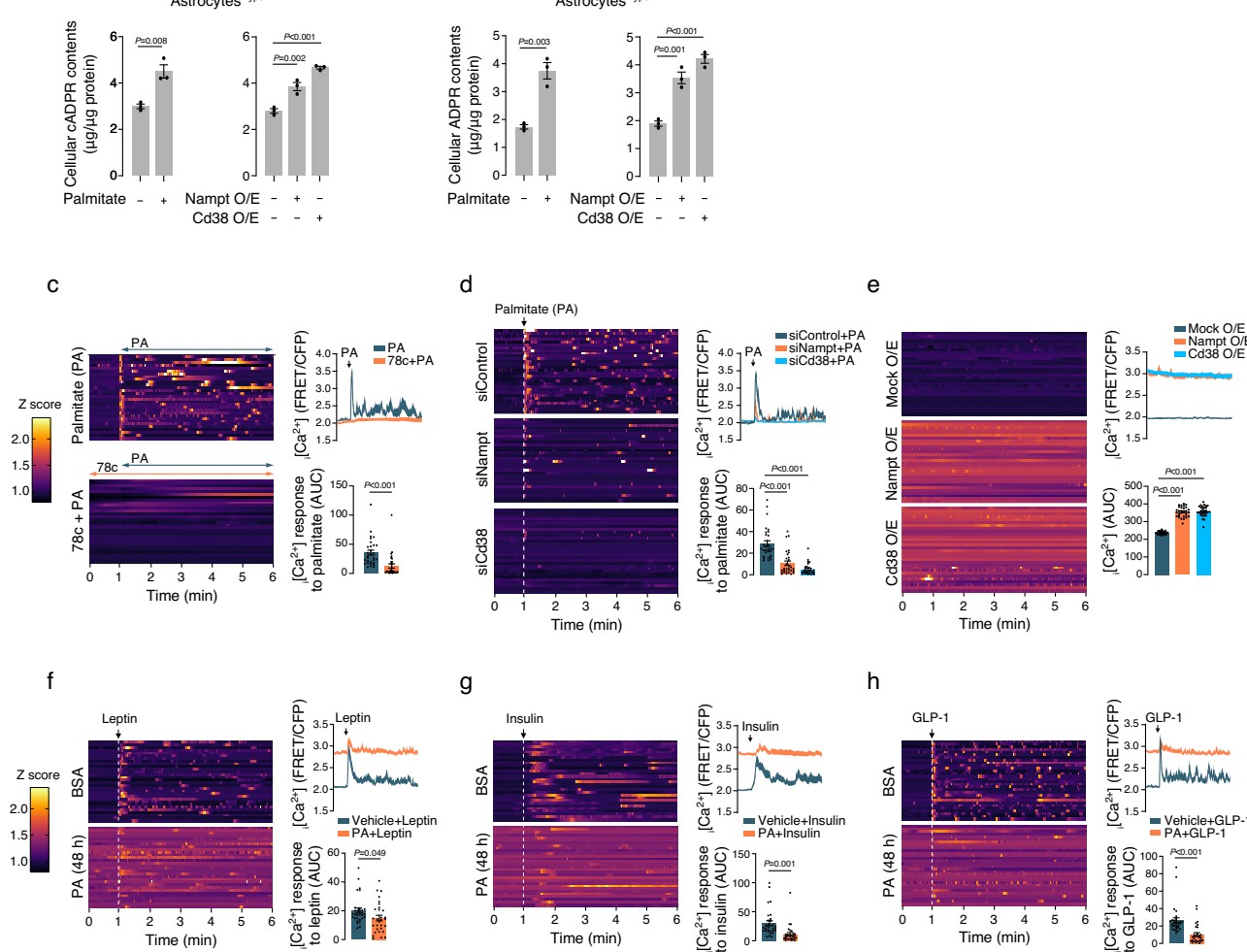

**Fig. 5 | Fat overload disrupts hypothalamic astrocyte Ca²⁺ signals via the NAMPT-NAD⁺-CD38 axis. a, b** Alterations in intracellular cADPR and ADPR contents in primary hypothalamic astrocytes treated with palmitate (200 μM, 48 h) or those overexpressing Nampt and Cd38 (n = 3 wells). **c, d** Effects of pre- (30 min) or co-treatment of 78c (100 μM) (**c**) or gene knockdown of Nampt or Cd38 (**d**) on palmitate (200 μM)-induced increase in intracellular Ca²⁺ (i[Ca²⁺]) levels in primary hypothalamic astrocytes (n = 30 cells). i[Ca²⁺] levels were assessed using a FRET-based Ca²⁺ sensor. The Ca²⁺ response was quantified as the area under the curve (AUC) of FRET/CFP values after palmitate treatment. **e** Significant increases in basal i[Ca²⁺] levels observed in primary hypothalamic astrocytes following Nampt or Cd38 overexpression (O/E) (n = 30 cells). **f–h** Decreased Ca²⁺ responses to treatment with leptin (1 μg/mL), insulin (0.01 unit/mL), or glucagon-like peptide-1 (GLP-1, 1 μg/mL) in primary hypothalamic astrocytes that were pre-exposed to palmitate (200 μM, 48 h) (n = 30 cells). The Ca²⁺ response was quantified as the AUC of the FRET/CFP values after hormone treatment. One-way ANOVA followed by Fisher's LSD test (**a, b** Nampt, Cd38 O/E, **d, e**) and two-sided unpaired t-test (**a, b**-palmitate, **c, f, g**). Three independent replicates were performed for all studies and measurement were taken from distinct samples. Results are presented as the mean *n* ± SEM. Source data are provided as a Source Data file.

fuel source over carbohydrates, as indicated by reduced RER, may underlie the lack of improvement in glucose tolerance and insulin resistance in AKO mice with reduced fat mass. Taken together, alterations in hypothalamic astrocyte activity and signaling can result in diverse metabolic outcomes.

Repeated exposure to saturated fatty acids triggers inflammatory responses in the hypothalamic ARH, which is characterized by the accumulation of activated microglia and reactive astrocytes[11,12,40]. Our study demonstrated that male Nampt-AKO mice exhibited lower levels of activated microglia and reactive astrocytes in the ARH, despite consuming an equal amount of HFD compared to WT controls. Furthermore, the expression of proinflammatory cytokines and chemokines was reduced in these mice. Consistently, activation of the NAD⁺ salvage pathway through Nampt overexpression or palmitate treatment stimulates proinflammatory gene expression and TNF-α secretion in hypothalamic astrocytes. These findings suggest a strong connection between the astrocytic NAD⁺ salvage pathway and hypothalamic inflammation in the context of HFD overconsumption.

It is highly plausible that during neuroinflammation, astrocytes communicate with neighboring microglia and neurons by releasing pro- and anti-inflammatory biomolecules[40,51]. Indeed, our study demonstrated that treatment with ACM collected from Nampt-overexpressing hypothalamic astrocytes upregulated the expression of proinflammatory genes in hypothalamic microglia and neurons. Moreover, blocking eNAMPT actions with a neutralizing antibody partially attenuated the proinflammatory effects of ACM. These results suggest that hypothalamic astrocytes with an activated NAD⁺ salvage pathway may trigger inflammation in adjacent neurons and microglia by releasing proinflammatory molecules, including eNAMPT. Considering that hypothalamic neurons elevated cellular NAD⁺ levels upon 6-h palmitate treatment, it is also possible that neurons release eNAMPT to elicit proinflammatory responses to surrounding astrocytes and microglia. Upon exposure to fatty acids, crosstalk between hypothalamic neurons, astrocytes, and microglia via temporally distinct NAD⁺ dynamics and eNAMPT secretion may contribute to hypothalamic proinflammatory responses.

The activation of the NAD$^+$ salvage pathway can induce both beneficial and detrimental outcomes, depending on the downstream NAD$^+$-dependent enzymes involved. Here, we identified CD38 as a downstream mediator of the inflammatory response induced by the NAD$^+$ salvage pathway in hypothalamic astrocytes. Palmitate treatment increased CD38 activity through the NAD$^+$ salvage pathway in cultured hypothalamic astrocytes. Hypothalamic CD38 activity was consistently elevated in mice chronically consuming a HFD. Furthermore, the administration of the chemical CD38 inhibitor 78c effectively suppressed HFD-induced hypothalamic inflammation and the progression of obesity. Consistently, depleting Cd38 expression specifically in hypothalamic astrocytes alleviated obesity by increasing EE and reducing the RER, mirroring the metabolic outcomes observed in Nampt-AKO mice. These findings support the hypothesis that astrocytic CD38 activation promotes DIO by operating downstream of the NAD$^+$ salvage pathway.

Elevated CD38 expression has been observed in CNS disorders associated with neuroinflammation, including aging, HIV infections, and autoimmune encephalitis[52]. Moreover, mice lacking CD38, a primary NADase in the brain, are protected from neurodegenerative and neuroinflammatory insults[53,54]. Given the neuroprotective and anti-inflammatory effects of NAD$^+$ [55,56], Cd38 depletion in hypothalamic astrocytes may improve hypothalamic inflammation by increasing NAD$^+$ levels. However, as palmitate-treated hypothalamic astrocytes showed elevated NAD$^+$ levels despite increased CD38 activity, it is plausible that NAD$^+$ production via the NAD$^+$ salvage pathway may outweigh CD38-induced NAD$^+$ degradation, at least under experimental conditions involving 48 h palmitate treatment of hypothalamic astrocytes. Another possibility is that CD38 activation may stimulate the transcription of proinflammatory genes through Ca$^{2+}$-dependent calcineurin/NFAT signaling, as reported in the experimental autoimmune encephalitis model[3]. This possibility is worth testing in future studies.

Spatiotemporally-regulated Ca$^{2+}$ oscillations play a significant role in astrocytic signaling in both physiological and pathological contexts. Astrocytes elevate i[Ca$^{2+}$] levels by releasing Ca$^{2+}$ from the endoplasmic reticulum (ER) through inositol triphosphate receptors (IP$_3$R) upon receptor stimulation[57]. In our study, acute palmitate treatment robustly increased i[Ca$^{2+}$] levels in hypothalamic astrocytes. The elevation in i[Ca$^{2+}$] levels induced by palmitate is mediated through mechanisms that depend on NAMPT and CD38. In vivo calcium studies in hypothalamic astrocytes of HFD-fed mice are necessary to validate these in vitro findings. As palmitate treatment as well as Nampt and Cd38 O/E increase cADPR levels in hypothalamic astrocytes, these treatments may elevate astrocytic i[Ca$^{2+}$] concentrations, either by releasing Ca$^{2+}$ from the ER via the ryanodine receptor (RyR) or by stimulating Ca$^{2+}$ influx, as reported previously[58]. Notably, chronic palmitate treatment inhibits astrocytic Ca$^{2+}$ responses to leptin, insulin, and GLP-1. Considering that palmitate increases i[Ca$^{2+}$] via NAMPT and CD38, the activated NAMPT−CD38 axis may inhibit hormonal Ca$^{2+}$ signaling in hypothalamic astrocytes. However, this does not exclude the possibility that various factors beyond the NAD$^+$ salvage pathway, such as reduced receptor availability, may also contribute to the blunted hormonal signaling. Aberrant activation of Ca$^{2+}$ signals in reactive astrocytes is closely associated with pathological alterations in gliotransmission, synaptic remodeling, and transcriptional programs[59]. Thus, the aberrant Ca$^{2+}$ signals triggered by excessive fat intake in hypothalamic reactive astrocytes may disrupt the normal interactions between astrocytes and neurons, further affecting the homeostatic regulation of energy balance.

In summary, our study provides valuable insights into the role of the NAD$^+$ salvage pathway−CD38 axis in hypothalamic astrocytes and its contribution to detrimental metabolic outcomes associated with excessive fat consumption. These findings underscore the potential of targeting this pathway as a therapeutic approach to combat obesity.

## Methods

### Ethical statement

All animal procedures were conducted in compliance with the guide for the care and use of laboratory animals (NIH) and approved by the Institutional Animal Care and Use Committee of the Asan Institute for Life Science (Seoul, Korea) under project license 2018-11-062.

### Cell culture

Astrocytes were isolated from the hypothalamus, cortex, and hippocampus of C57BL/6J (C57) mouse pups at postnatal 1−2 days and subjected to primary culture following established protocols[60]. In brief, brain tissues from different brain regions were dissociated into single cells by mechanical chopping and filtration using a 40-μm mesh. Subsequently, cells were cultured in high-glucose Dulbecco's Modified Eagle's Medium (DMEM, Thermo Fisher Scientific) supplemented with 10% fetal bovine serum (FBS; Gibco, 26140079) and 1% penicillin/streptomycin (Gibco, 15140122). To eliminate microglia and oligodendrocytes, culture flasks were shaken at 240 rpm overnight, and the supernatant was discarded every 2 days. For the primary culture of hypothalamic microglia, cells were obtained from the hypothalamic region of C57 neonates at postnatal days 1−2 and cultured in the same medium used for astrocyte culture. Upon reaching confluence, microglia were collected by shaking at 180 rpm for 30 min and then seeded onto a Poly-D-lysine-coated culture plate. For the primary culture of hypothalamic neurons, cells were isolated from prenatal rat embryos at embryonic day 18. Briefly, brain tissues were dissociated into single cells through mechanical chopping and 0.01% trypsin-EDTA treatment for 15 min. Subsequently, Neurobasal (NB) media (Gibco, A35829) with 10% FBS was added to cells, followed by centrifugation. Cells were seeded onto a poly-D-lysine-coated culture plate using NB media supplemented with 1% B27 (Gibco, A35828), 2 mM L-glutamine, and 1% penicillin/streptomycin[61]. N1 hypothalamic neuron cells were obtained from Cedarlane (CED-CLU101) and cultured in DMEM supplemented with 10% FBS and 1% penicillin/streptomycin.

### Animals

C57 male mice and female Sprague Dawley pregnant rats aged 8 to 10 weeks were obtained from Orient Bio. Nampt-AKO mice were generated by crossbreeding Tg (GFAP-cre/ERT2) 13Kdmc mice (Mouse Genome Institute, #3712447) with Nampt-floxed mice (Jackson Laboratory, #034242). Gene deletion was induced by intraperitoneal injection of tamoxifen (100 mg/kg dissolved in corn oil with 10% ethanol) for 5 consecutive days after the mice had reached 7 weeks of age. All animals were housed under controlled environmental conditions, including a temperature of 22 ± 1 °C, humidity maintained at 55% ± 5%, and a 12-h/12-h light/dark cycle, with lights on from 8 a.m. until 8 p.m. The mice had free access to a standard chow diet (containing 12.5% of calories from fat; Cargill Agri Purina, 38057) and water unless otherwise specified. To induce DIO, mice were fed an HFD (containing 60% fat content, Research Diets, D12492). Mice were euthanized in a carbon dioxide chamber at the designated time points during the study.

### Mouse metabolic phenotyping

Weekly monitoring of body weights and food intake was conducted in both male and female AKO and WT mice between 9:00 a.m. and 10:00 a.m. using a digital weight scale after 5 weeks. Energy expenditure was measured at the indicated ages using a comprehensive laboratory animal monitoring system (CLAMS, Columbia Instrument). Mice were placed in the CLAMS chambers for a total of 3 days; the first day served as an adaptation period for the mice, while data from the second and third days were used for analysis. The average EE values on the second and third days were utilized for ANCOVA analysis. Lean mass and fat mass were measured using dual X-ray absorptiometry (Lunar PIXImus). Visceral and subcutaneous fat mass were determined

using magnetic resonance imaging (Agilent Technology). For the glucose tolerance test, D-glucose (1 g/kg, Sigma) was orally administered to mice under overnight fasting conditions. For the insulin tolerance test, insulin (Humulin-R® 0.25 U/kg, Eli Lilly) was injected into the peritoneum of overnight fasted mice. Blood samples were obtained from the tail vein immediately before and 30, 60, 90, and 120 min after injections to measure blood glucose levels using a glucometer (ACCU-CHEK®, Aviva Plus System).

## Preparation of fatty acid solution

To prepare the fatty acid solution, 50 mg of sodium palmitate (Sigma, P9767) was dissolved in 10 mL of 0.02 M NaOH and added to 10% BSA solution at a 1:3 v/v ratio to prepare a 5 mM BSA-conjugated palmitate stock solution. Oleic acid (2.82 mg, Sigma, O1257) was dissolved in 1 mL of sterile water to prepare a 10 mM oleic acid stock solution. The final concentrations of palmitate and oleate solutions were obtained by diluting their respective stock solutions with a culture medium.

## Measurement of NAD⁺ and NADH contents

$NAD^+$ and NADH levels were measured using an $NAD^+$/NADH assay kit (Bioassay systems, E2ND-100). Homogenized tissue or cell samples were lysed with the $NAD^+$ or NADH extraction buffer included in the kit and then subjected to heat extraction at 60 °C for 5 min. After centrifugation of the samples, 40 µl of supernatant was used for $NAD^+$ or NADH analysis. The $NAD^+$ and NADH contents were then normalized based on the protein content.

## Enzyme activity assay

To assess cellular NAMPT activity, cells were washed twice with phosphate-buffered saline (PBS) and then lysed using ice-cold cell lysis buffer containing 20 mM Tris (pH 8.0), 250 mM NaCl, 1 mM EDTA, 1 mM EGTA, 1% Triton X-100, 1 mM DTT, and protease inhibitors. After centrifugation, 5 µl of the supernatant was loaded onto a 96-well plate with the NAMPT enzymatic activity reaction mixture, following the manufacturer's protocol (Abcam, ab221819). Enzyme activity was normalized on the basis of the amount of protein, and all samples were assayed in duplicate.

To assay CD38 activity, tissue or cell samples were sonicated in a buffer containing 40 mM Tris and 250 mM sucrose. Following centrifugation, 50 µg of the protein sample was loaded onto a 96-well plate along with 100 µl of the CD38 enzymatic activity reaction mixture, including BSA (40 mg/ml), nicotinamide guanine dinucleotide (NGD, 200 µM), and nicotinamide 1, N6-ethenoadenine dinucleotide (ε-NAD, 50 µM), all in a 40 mM Tris, 250 mM sucrose buffer, as previously described[62]. CD38 activity was determined by monitoring the change in fluorescence at 300 nm excitation and 410 nm emission. Fluorescence readings were normalized based on protein content.

## Immunoblotting

Primary hypothalamic astrocyte lysates (50 µg protein) were separated by 10% SDS-PAGE and transferred to a PVDF membrane (GE Healthcare). Following incubation in blocking buffer, the membranes were incubated overnight at 4 °C with an antibody against NAMPT (1:1500, mouse; Adipogen, AG-20A-0034). The blots were developed using a densitometer (VersaDoc Multi Imaging Analyzer System, Bio-Rad) and normalized using β-actin immunoblotting (1:1000, mouse; Santa Cruz, sc-47778).

## Real-time PCR

Total RNA was extracted from cells and tissues using TRIzol reagent (Invitrogen) according to the manufacturer's protocol. Subsequently, 1 µg of RNA was reverse transcribed to generate cDNA using M-MLV reverse transcriptase (Invitrogen). Next, real-time PCR (QuantStudio 5, Applied Biosystems) was employed to assess the mRNA levels of genes using the corresponding primers (Supplementary Table. 1). The expression level of each mRNA was normalized to that of glyceraldehyde 3-phosphate dehydrogenase (Gapdh) or ß-actin mRNA.

## TNF-α secretion

Mouse primary hypothalamic astrocytes were seeded in 6-well plates and treated with chemicals and/or transfected with Nampt-expressing plasmids on the following day. After 24 h, the medium was replaced, and cells were cultured for an additional 24 h. Subsequently, the medium was harvested, and TNF-α secretion from primary cultured astrocytes was determined by measuring the TNF-α levels in the medium using a mouse TNF-α Quantikine ELISA kit (R&D Systems, MTA00B) and ELISA assay reader (Tecan, Infinity 200 PRO).

## Gene overexpression and knockdown

Plasmids expressing Nampt (obtained from Dr. Shin-ichiro Imai at Washing University) and Cd38 (Sino Biological, MG50191-NY) were transfected into cells at the indicated doses using Lipofectamine 3000 (Invitrogen) when they reached 60% ~ 70% confluency. Small interfering RNA (siRNA) targeting murine Nampt (Dharmacon, E-040272-00-0005), Cd38 (Thermo Fisher Scientific, 160087), Sirt1 (Dharmacon, M-049440-00), Sirt2 (Dharmacon, M-061727-01), and Sirt6 (Dharmacon, M-013306-02), along with non-targeting scrambled siRNA, were used. Cells were plated and transfected with siRNAs (100 pmol) using Lipofectamine. Successful gene overexpression or knockdown was confirmed by real-time PCR of cells 48 h after transfection.

## Hypothalamus immunostaining

Mice were anesthetized by intraperitoneal injection of 40 mg/kg Zoletil® and 5 mg/kg Rompun®. Subsequently, they were perfused with 50 mL saline, followed by 50 mL of 4% paraformaldehyde (PFA) via the left ventricle of the heart. Whole brains were collected, fixed with 4% PFA for 24 h, and then dehydrated in 30% sucrose solution until the brains sank to the bottom of the container. Coronal brain sections, including the hypothalamus, were sliced to a thickness of 30 µm using a cryostat (Leica, Wetzlar, Germany). One out of every five slices was collected and stored at −80 °C. For staining, hypothalamic slices were permeabilized in PBS with 0.5% Tween 20 or Tris-buffered saline with 0.5% Tween 20 (for TMEM119) for 5 min, followed by blocking with either 5% normal donkey serum, normal goat serum (for IBA1), or bovine serum albumin (for TMEM119) at room temperature for 1 h. Subsequently, the slices were incubated overnight at 4 °C with primary antibodies targeting the following proteins: GFAP (1:1000, rabbit; Millipore, ab5804), S100β (1:1000, rabbit; Abcam, ab51642), IBA1 (1:1000, goat; Abcam, ab5076), TMEM119 (1:1000, rabbit; Abcam, ab209064), CD38 (1:1000, mouse; Novus biologicals, NBP2-25250), C3 (1:1000, rat; Abcam, ab11862), p-STAT3 (1:1000, rabbit; Cell signaling, 9131), and c-Fos (1:1000, rabbit; SYSY, 226008). After washing, the slides were incubated with Alexa Fluor 488-, 555-, or 633-conjugated secondary antibodies (1:1000) at room temperature for 1 h. For nuclear staining, the slides were counterstained with DAPI (1:10000) for 5 min before mounting. Immunofluorescence images were captured using an LSM 710 confocal microscope (Carl Zeiss). Quantification of fluorescence intensity and cell counting were performed using the ZEN microscope software (version 2.1 blue edition, Carl Zeiss). Approximately four brain slices from each animal, including the hypothalamic ARH, were examined for each staining analysis. The average cell number and fluorescence intensity in the ARH per brain slice are presented.

## In situ hybridization

Whole brains were collected from mice after cardiac perfusion with 50 mL of saline, followed by 50 mL of 4% PFA in diethyl pyrocarbonate (DEPC) water through the left ventricle under anesthesia with 40 mg/

kg Zoletil® and 5 mg/kg Rompun®. Subsequently, the collected brains were post-fixed with 4% PFA at 4 °C overnight and then stored in a 30% sucrose solution until the brains sank to the bottom of the container. The brain was sectioned into 14-μm-thick slices using a cryostat (Leica). The brain slices were subjected to antigen retrieval and protease agent, followed by hybridization with Nampt probes (ACDBio, 413491) for 2 h at 40 °C, with step-wise amplification according to the manufacturer's protocol. Following RNA in situ hybridization, the brain sections were subjected to immunofluorescence staining. Briefly, the sections were incubated with 3% BSA in PBS for 1 h, followed by overnight incubation with a diluted GFAP antibody (1:1000, rabbit; Millipore, ab5804) in 3% BSA in PBS at 4 °C. Subsequently, the slides were washed and incubated with the appropriate Alexa Fluor 488-conjugated secondary antibodies (1:1000) at room temperature for 1 h. We also performed Pomc in situ hybridization using Pomc probes (ACDBio, 314081), followed by c-Fos immunostaining using the aforementioned method. Immunofluorescence images were acquired using a confocal microscope (Carl Zeiss 710, Germany). Approximately four brain slices from each animal, including the hypothalamic ARH, were examined. The average Nampt fluorescence intensity in astrocytes in the hypothalamic ARH, hippocampal CA1 region, and temporal cortex or the percentages of c-Fos$^+$ cells among ARH POMC neurons per brain slice are presented.

## Adipose tissue histology and immunostaining

Mice were perfused with 4% PFA via the left ventricle as described in the immunostaining section. The BAT, iWAT, and gWAT were then dissected, post-fixed in 4% PFA overnight, and embedded in paraffin. Paraffin-embedded sections were stained with hematoxylin and eosin (H&E) and examined under a microscope (Olympus, BX53, Japan). Adipocyte sizes were quantified using the Image J program (NIH). Sympathetic nerve terminals were stained using a previously established method[63]. Paraffin sections of mouse adipose tissue were deparaffinized and dehydrated through sequential washing: twice for 10 min in xylene, followed by washes in 100%, 95%, and 70% ethanol (2 min each, repeated twice). The sections were then rinsed in distilled water and placed in preheated citrate buffer (0.01 M, pH 6.0) for 5 min, followed by immersion in hot citrate buffer (58 °C) and cooling to room temperature. Subsequently, the sections were washed three times with Tris-buffered saline (TBS) for 5 min and then transferred into methanol solutions containing 0.5% $H_2O_2$, followed by another TBS wash. To prevent non-specific binding, the sections were pretreated by blocking with 2% normal horse serum for 1 h at room temperature. Following the blocking step, the slides were incubated overnight at 4 °C with rabbit anti-tyrosine hydroxylase (TH) antibody (1:250, rabbit; MerckMillipore, ab152). Subsequently, the slides were treated with Alexa Fluor 488-conjugated anti-rabbit secondary antibody (1:500) for 1 h at room temperature. Immunofluorescence images were captured using a confocal microscope (Carl Zeiss 710, Germany) to visualize sympathetic nerve terminals in adipose tissues. Adipose tissue sympathetic innervation was assessed by measuring the fluorescence intensity of TH per mm$^2$.

## Astrocyte isolation

Astrocytes from MBH, cortex, and hippocampus were isolated using magnetic-activated cell sorting (MACS). Briefly, mice were euthanized by decapitation. Tissue blocks were swiftly collected, finely chopped into small pieces in ice-cold Dulbecco's phosphate buffered saline, and then transferred to gentleMACS™ C tubes. Subsequently, the tissues were dissociated into single-cell suspensions using the Adult Brain Dissociation kit (Miltenyi Biotec, 130107677) and the gentleMACS™ Octo Dissociator (Miltenyi Biotec). After centrifugation, the cells were resuspended and labeled with an anti-ACSA-2 microbead kit (Miltenyi Biotec, 130097678) for

astrocyte isolation. To validate the accurate isolation of region-specific astrocytes, we performed qPCR analysis on brain region-specific astrocyte markers such as Agt, Lhx2, and Emx2[64] (Supplementary Fig. 9).

## Astrocyte–microglia or astrocyte–neuron interaction study

To investigate the impact of Nampt overexpression in hypothalamic astrocytes on hypothalamic microglia and neurons, we collected astrocyte-conditioned medium from hypothalamic astrocytes overexpressing Nampt, and then lyophilized and reconstituted it in 1 mL of DMEM. Next, the reconstituted medium was used to treat primary cultured hypothalamic microglia and neurons. In the control group, an astrocyte-conditioned medium was collected from hypothalamic astrocytes overexpressing the mock plasmid. After 48 h of treatment, microglia and neurons were harvested for gene expression analysis.

## Hypothalamic AAV injection

To knock down the Nampt or Cd38 gene in hypothalamic astrocytes, AAV-DIO-shNampt-eGFP (shNampt-AAV) and its corresponding control, AAV-DIO-shControl-eGFP (shControl-AAV), as well as AAV-DIO-loxP-CMV-mCherry-STOP-loxP-shCD38 (shCd38-AAV) and its corresponding control, AAV-DIO-loxP-CMV-mCherry-STOP-loxP-shControl (shControl-AAV), were generated at the Virus Core of the Korean Institute for Science and Technology. The AAV constructs, at a concentration of $1 \times 10^{13}$/ml, were microinjected bilaterally into the ARH of 10-week-old hGFAP-Cre$^{ERT2}$ male mice that had received tamoxifen injections 2 weeks prior. The injections were conducted at a depth of 5.7 mm, 1.6 mm caudal on the bregma, and 0.1 mm lateral to the sagittal suture. On each side, a 0.5 μl volume of the construct was delivered using a syringe pump (Harvard Apparatus) at a rate of 100 nl/min for 5 min[61]. Two weeks after AAV injection, the mice were fed a HFD, and the body weights were measured weekly until the mice were sacrificed. The EE and RER were assessed 2 weeks after the initiation of HFD feeding, while the body mass was measured at 2 and 4 weeks after HFD feeding and 4-week data are presented. At the end of the study, the success of viral injection was validated by observing GFP expression (for shNampt-AAV study) or deletion of mCherry (for shCd38-AAV study) in ARH astrocytes. Gene knockdown was confirmed by the reduction in Nampt or Cd38 expression in ARH astrocytes compared to that in the control groups. Animals that did not exhibit successful AAV injections in bilateral ARH astrocytes were excluded from the data analysis.

## Intracerebroventricular infusion of 78c

Stainless steel cannulas (26 gauge; Plastics One) were implanted into the third cerebroventricle of 8-week-old C57 mice. The coordinates for implantation were 1.5 mm caudal to bregma and 5.0 mm ventral to the sagittal sinus[61]. The procedure was performed using stereotaxic surgery under anesthesia with 40 mg/kg Zoletil® and 5 mg/kg Rompun®. Subsequently, an osmotic minipump (Alzet, 1004) was placed in the interscapular area and connected to the i.c.v.-implanted cannula via polyurethane tubing. Briefly, 78c, dissolved in 0.9% saline solution, was infused into the cerebroventricle over 4 weeks, at a rate of 2.8 μg per day, via the osmotic pump-implanted cannula. After surgery, the mice were fed a HFD and their body weights were monitored weekly. The EE and RER were assessed on treatment day 25 using the CLAMS. The body mass was determined 1 day before CLAMS assessment. At the end of 78c treatment, the mice were sacrificed to collect the hypothalamus and brown and white adipose tissues. Successful infusion of 78c was confirmed by weighing the osmotic pumps after sacrifice, while the proper location of the ICV cannula was confirmed by black ink injection via the cannula before sacrifice. Data from animals showing a misplaced cannula and malfunctioning osmotic pumps were excluded from analysis.

## Leptin injection study

Leptin (5 mg/kg) was administered into the peritoneum in mice following a 5-h fast in both Nampt-WT and Nampt-AKO male mice fed a HFD for 2 weeks. Thirty minutes after leptin injection, the mice were subjected to transcardiac perfusion with 4% PFA under anesthesia, as detailed in the hypothalamus immunostaining section, before collecting brain tissues for p-STAT3 staining.

## Cyclic ADPR and ADPR assay

Mouse primary hypothalamic astrocytes were seeded in 6-well plates and treated with palmitate (200 μM, 48 h) or transfected with Nampt- or Cd38-expressing plasmids for 48 h. Subsequently, the cells were scraped, and the cellular contents of cyclic ADPR (cADPR) and ADP-ribose (ADPR) were quantified using the cADPR ELISA kit (Mybiosource, MBS3806816) and the ADPR ELISA kit (Mybiosource, MBS169573), respectively, following the manufacturer's protocol.

## Astrocyte calcium imaging

Hypothalamic astrocytes were cultured in confocal dishes and 48 h prior to imaging, cells were transfected with a plasmid expressing the fluorescent resonance energy transfer (FRET)-based $Ca^{2+}$ sensor YC3.60[65]. To prepare for calcium imaging, the cells were switched to calcium- and phenol red-free HBSS media (Gibco, 14175) 30 min before microscopic examination. Imaging was conducted using a Nikon Ti-E inverted microscope (Nikon) equipped with a perfect focus system (PFS) and an iXon Ultra 897 EMCCD camera (Andor, UK). Excitation and emission filter wheels were employed for image capture. The imaging process used 2 × 2 binning mode and a 200-ms exposure time. For dual-emission ratio imaging of the intramolecular FRET probe, separate images were acquired for the cyan fluorescent protein (CFP) and FRET channels. All acquired images were processed and analyzed using MetaMorph software (Universal Imaging, Downingtown, PA, USA). The necessary components for the dual-emission ratio imaging setup, including an FF01-438/24-25 excitation filter, an FF458-Di02-25×36 dichroic mirror, and FF01-483/32-25 CFP and FF01-542/27-25 FRET emission filters, were purchased from Semrock (Rochester, NY, USA).

## Cell viability test

The cell viability assay was conducted using a CCK-8 kit (Dojindo, #CK04) following the manufacturer's instructions. Briefly, N1 cells were seeded in 96-well plates and allowed to grow overnight. Subsequently, the cells were exposed to palmitate solution at concentrations of 100–1000 μM for 48 h, followed by incubation with CCK-8 solution for 3 h. Finally, absorbance readings were obtained at 450 nm using a microplate reader (Tecan, Switzerland).

## Statistical analysis

Statistical analysis was performed using Prism software version 8.0 (GraphPad) or SPSS version 24 (IBM Analytics). Statistical significance among the groups was assessed using one- or two-way analysis of variance (ANOVA) followed by Tukey's LSD post hoc analysis or a two-sided unpaired Student's t-test, as appropriate. The comparison of EE between the groups was analyzed using one-way ANCOVA. Statistical significance was defined at $P < 0.05$. Data are presented as the mean ± standard error of the mean (SEM).

## Reporting summary

Further information on research design is available in the Nature Portfolio Reporting Summary linked to this article.

## Data availability

All data generated or analyzed during this study are provided with this paper in the source data files. Source data are provided with this paper.

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

## Acknowledgements

This study was supported by grants from the National Research Foundation of Korea (NRF) funded by the Ministry of Science and ICT of Korea (2020R1A2C3004843 and 2022M3E5E8017213 to M.S.K, 2022R1C1C1012590 to S.H.M, and 2022R1C1C2007378 to G.M.K).

## Author contributions

J.W.P. and M.S.K. were involved in the experimental design; J.W.P., S.E.P., W.H.K., W.H.J., J.H.C., E.R., G.M.K., S.J.K., H.S.L., C.B.P., S.Y.J., S.Y.M. performed the experiments; C.H.L., S.Y.K., H.J.C., S.H.M. and C.J.L. discussed the data; J.W.P. and M.S.K. analyzed the data and wrote the manuscript; M.S.K. directed the study.

## Competing interests

The authors declare no competing interests.
