## [Peer Review File · Nature Communications]

Hypothalamic astrocyte NAD⁺ salvage pathway mediates the coupling of dietary fat overconsumption with obesity in a mouse model of obesityREVIEWER COMMENTS

Reviewer #1 (Remarks to the Author):

The authors conducted several systematic in vitro and in vivo experiments to show that the hypothalamic-astrocyte-specific NAD⁺ salvage pathway is required for fatty acid induced hypothalamic inflammation and HFD-induced weight gain in mice - to ultimately uncover the mechanisms underlying astrocytic involvement in the progression of obesity. Upon confirmation of the involvement of astrocytic NAMPT, authors demonstrated that NAMPT-NAD⁺ acts via CD38 to induce neuroinflammation and weight gain in HFD-fed mice. Finally, authors demonstrated that palmitate-induced NAMPT inhibition leads to impairment in basal and hormone-induced CA2⁺ levels, ultimately leading to astrocyte dysfunction.

The manuscript is well written and contributes to the understanding of how astrocytes are involved in hypothalamic energy homeostasis. Experiments were generally well designed; however, further controls and discussion of specific findings would enhance the manuscript greatly.

Methods

- In mHypoE-46 hypothalamic cells and primary hypothalamic cultures, 50 uM palmitate treatment for 24 h increased NAMPT levels (reference #44) as indicated by the authors in the discussion. The authors attribute this difference to potential dual responses by neurons. However, the concentration of palmitate used (200 uM) is high. Please describe how palmitate was prepared and dissolved. Discuss whether cell viability was affected at later timepoints and whether this may be the reason for the decrease in Naprt, Nmnat and NAD⁺ in neurons at later time points in this study (Fig 1A).

- Indicate source of N1 cells

Results/Discussion

- In almost all experiments where an inhibitor was used (FK866, siNAMPT, 78c, siCD38, Sirtinol, Olaparib), no inhibitor only/siRNA only control is shown. If basal levels of the measured factor (i.e basal NAD⁺ or basal inflammatory markers) are decreased by the inhibitor alone compared to vehicle alone, we do not know whether the inhibitor actually blocked the effect of the treatment or whether it purely decreased basal levels. Did authors perform these controls?

- Was any screening done for the astrocytes isolated from the cortex/MBH/hippocampus to verify region-specificity?

- Fig 2C - It would be meaningful to show how cellular NAD⁺ content is affected by the KO. Although Nampt levels are mitigated, how was the activity and the resulting NAD⁺ levels affected? In other words, how potent is the KO in terms of affecting NAD⁺?

- Fig 2D - Were control groups taken out to 16 weeks as well? How significant is the KO weight loss physiologically?

- Fig 2K/L - Along the same point, if the KO mice didn't improve insulin sensitivity or glucose tolerance, can the authors comment on the physiological relevance of the weight loss seen? Perhaps the HFD-mice were not taken out long enough to see an impairment in glucose tolerance/insulin resistance in the WT mice, and as such, no benefit could have been seen in the KO mice.

- Fig 3D - typo in legend (STAT3 not START3). Did authors observe differences in leptin receptor levels as well? A mechanism for this increased STAT3 signaling upon NAMPT deletion would be useful to explore/discuss.

- Fig 3H - In both neurons and microglia, the ACM-NAMPT O/E increases pro inflammatory cytokines that aren't all completely blocked by the neutralizing antibody. This could be due to the

potency of the neutralizing antibody OR the fact that the effects are not all directly attributable to eNAMPT. To address this, looking at NAD⁺ content in these experiments would be beneficial, if feasible. Alternatively, authors can discuss the potency of the antibody if known from other studies.

- Fig 4D – What happens to CD38 activity in isolated MBH astrocytes from NAMPT KO animals fed a HFD (i.e samples from exp in 2D, only WT is shown in 4D)? Wouldn't this further reinforce that NAMPT-NAD⁺ acts via CD38?
- Fig 4G – Comment on why the body weight gain of these mice is less than the ones in 2D on just the HFD alone?
- Discussion - "In line with this, activation of the NAD⁺ salvage pathway through Nampt overexpression or palmitate treatment stimulates proinflammatory gene expression and TNF- α secretion in hypothalamic astrocytes." From Fig 3F - palmitate has inflammatory actions that are not mediated by the NAMPT pathway as FK866 does not completely block the palmitate-mediated increase in IL6 and MCP-1. Authors should acknowledge this.

Overall this is a nice study that would benefit from some important controls in order to gain confidence in the results.

Reviewer #2 (Remarks to the Author):

Overall the study is well done, presented and discussed. We have the few comments mentioned below:

1. An assay directly measuring changes in CD38 levels or activity in hypothalamic astrocytes when NAMPT is manipulated (either knocked out with the hGFAP-Cre ERT2 or overexpressed in the Nampt O/E astrocytes) would better support the conclusion that it is acting downstream of the NAD⁺ salvage pathway, as stated in the abstract.
2. Measuring of hypothalamic astrocyte NAD⁺ levels in the AKO mouse would rule out any compensatory increase in press- handler and de-novo synthesis pathways that can produce NAD⁺ and that may be occurring an in-vivo setting.
3. It would be better to match the mice for the leptin treatment experiment in Figure 3D for adiposity (fat mass) rather than just body weight (Supp figure 2A) as this would be a better control for the potential confounding effects of this parameter in leptin sensitivity.

Reviewer #3 (Remarks to the Author):

Reviewer #4 (Remarks to the Author):

The present work addresses the timely topic of high fat diet induced astrocyte reactivity and inflammation in the hypothalamus. Employing a combination of in vitro and in vivo techniques, the authors shed light on the role of the NAD⁺ salvage pathway and its downstream component, CD38, in relation to obesity-related metabolic aspects.

In their initial investigation, utilizing primary cultured hypothalamic astrocytes, the authors employed pharmacological inhibitors to convincingly establish the critical importance of the salvage pathway in driving the increase in NAD⁺ levels induced by palmitate in astrocytes. Subsequently, they generated mice with astrocyte-specific NAMPT depletion to target the NAD⁺ pathway and assess its implications in vivo. This experiment revealed that in the absence of an activated NAD⁺ salvage pathway, mice subjected to a high-fat diet exhibited reduced adipose tissue accumulation and displayed enhanced energy expenditure, despite having a similar caloric intake. These changes were associated with a decreased level of inflammatory processes in the arcuate nucleus, and these findings were corroborated using an alternative model involving shRNA injection in the arcuate nucleus.

Furthermore, an examination of downstream NAD⁺-dependent enzymes in primary astrocyte cultures revealed that inhibiting SIRT1/2, did not alter the proinflammatory effects associated with the overexpression of Nampt. On the contrary, CD38blockade recapitulated the beneficial effects of NAMPT depletion in high fat diet fed animals.

Well controlled and synergistic approaches for targeting key components of the NAD⁺ pathway have been developed using transgenic mice, viral tools and pharmacology in vivo combined with metabolic assessments.

The manuscript is clearly written and organized. Here are my comments.

Majors:

1/One limitation of this study is to generalize the results derived from cultured astrocytes or from the large region of the mediobasal hypothalamus (MBH) to the entire hypothalamus. This limitation arises from the fact that astrocytes within different nuclei of the hypothalamus can exhibit significant functional variability and distinct responses when exposed to a high-fat diet (HFD). My recommendation is that the authors discuss this important limitation.

2/Page 24: regarding the section on hypothalamic AAV injections. In their approach to target the ASH, the authors inject a substantial volume of 500 μ l of virus on each side that could lead to a larger infection than just the ASH. Could the author provide a figure showing the extent of the viral infection?

3/What was the rationale behind the authors selecting a 4-week exposure to a high-fat diet (HFD) in their experimental design? It would be insightful to understand whether it is based on metabolic considerations.

4/Even if the article is centered on astrocytes, it is interesting to notice the differential effect of palmitate on neurons and astrocytes since most of the literature is on NAD⁺ pathway in neurons. In Figure 1, it is observed that neurons exhibit a more rapid increase in NAD⁺ levels compared to astrocytes. This temporal distinction in NAD⁺ dynamics within these two cell types raises the question of the crosstalk and signaling between neurons and astrocytes. The interplay between neurons and astrocytes might impact NAD⁺ dynamics. All the culture experiments are astrocytes only. Could the authors discuss this point?

5/Line 146, the authors mentioned that the effect was absent in females AKO mice. Did the authors tested females on the different group of mice studied? This is not mentioned in the methods.

6/It would be beneficial to provide a more detailed description of the metabolic experiments of the study. Figures displays 2 days in the metabolic cages, and these 2 days appears quite different for EE (figure 2h). How long did the run last? Are the traces averaged over multiple days or not? Were the statistics performed on 2 days with dark and light phases together?

7/In figure 4g and h, body weight gain and fat mass are decreased in 78c injected mice, could the

authors provide the graph of the lean mass in h also? In figure 4h, is the regression significant for the 78c injected group?

8/In this specific group of mice, the authors have not supplied information regarding the locomotor activity. Could the variations in locomotor activity in the 78c mice be a contributing factor to the elevated energy expenditure in these particular mice?

9/The rationale of the latest experiment is not fully clear to me. The authors demonstrate that leptin, insulin, and GLP-1 induce a calcium increase in hypothalamic astrocytes, which is subsequently affected by exposure to palmitate. Nevertheless, it's crucial to consider that various factors beyond the NAD⁺ salvage pathway, such as receptor availability, could potentially contribute to this observed effect.

Minor comments :

In the methods section Alexa-Fluor is misspelled as Alexa-Flour

Line 327 : "it has been shown" instead of "it have been shown".

Line 587 : 78C the c is missing. In the same paragraph, there are multiple spelling mistakes

Reviewer #5 (Remarks to the Author):

Reviewer's comments

We wish to express our sincere gratitude to the reviewers for their valuable comments. We believe that their advice has greatly enhanced the quality of our manuscript.

Reviewer #1 (Remarks to the Author):

The authors conducted several systematic in vitro and in vivo experiments to show that the hypothalamic-astrocyte-specific NAD⁺ salvage pathway is required for fatty acid induced hypothalamic inflammation and HFD-induced weight gain in mice - to ultimately uncover the mechanisms underlying astrocytic involvement in the progression of obesity. Upon confirmation of the involvement of astrocytic NAMPT, authors demonstrated that NAMPT-NAD⁺ acts via CD38 to induce neuroinflammation and weight gain in HFD-fed mice. Finally, authors demonstrated that palmitate-induced NAMPT inhibition leads to impairment in basal and hormone-induced CA2⁺ levels, ultimately leading to astrocyte dysfunction.

The manuscript is well written and contributes to the understanding of how astrocytes are involved in hypothalamic energy homeostasis. Experiments were generally well designed; however, further controls and discussion of specific findings would enhance the manuscript greatly.

Methods

- In mHypoE-46 hypothalamic cells and primary hypothalamic cultures, 50 μ M palmitate treatment for 24 h increased NAMPT levels (reference #44) as indicated by the authors in the discussion. The authors attribute this difference to potential dual responses by neurons. However, the concentration of palmitate used (200 μ M) is high. Please describe how palmitate was prepared and dissolved. Discuss whether cell viability was affected at later timepoints and whether this may be the reason for the decrease in Naprt, Nmnat and NAD⁺ in neurons at later time points in this study (Fig 1A).

Response: To investigate whether treatment with high concentrations of palmitate induces neuronal cell death, we conducted an experiment in which N1 cells were exposed to palmitate at concentrations of 100–1000 μ M for 48 h. Following the treatment period, we assessed cell viability. Our results indicated a significant reduction in cellular viability at palmitate concentrations exceeding 750 μ M, whereas 200 μ M of palmitate, as utilized in our study, showed no impact on cell viability (Supplementary Fig. 1, Line 79). Therefore, the observed decrease in Naprt, Nmnat, and NAD⁺ levels in hypothalamic neurons, treated with 200 μ M palmitate for 48 h, did not arise from cell death. Additionally, the Methods section provides details on the preparation of palmitate and oleate solutions (refer to line 478–483).

- Indicate source of N1 cells

Response: We have included the source of N1 cells in the Methods section (refer to Line 445).

Results/Discussion:

- In almost all experiments where an inhibitor was used (FK866, siNAMPT, 78c, siCD38, Sirtinol, Olaparib), no inhibitor only/siRNA only control is shown. If basal levels of the measured factor (i.e basal NAD⁺ or basal inflammatory markers) are decreased by the inhibitor alone compared to vehicle alone, we do not know whether the inhibitor actually blocked the effect of the treatment or whether it purely decreased basal levels. Did authors perform these controls?

Response: While our *in vitro* studies initially included inhibitor-alone treatment groups, the corresponding data were not presented. We have now incorporated the data from inhibitor-alone treatment groups into Figures 1g, h, 4a–c, and Supplementary Figures 6a–e.

- Was any screening done for the astrocytes isolated from the cortex/MBH/hippocampus to verify region-specificity?

Response: We performed a screening test in astrocytes isolated from the cortex, MBH, and hippocampus to verify the region specificity by qPCR analysis of region-specific astrocyte markers such as *Agt*, *Lhx2*, and *Emx2*. The data are presented in Supplemental Figure 9.

- Fig 2C - It would be meaningful to show how cellular NAD⁺ content is affected by the KO. Although *Nampt* levels are mitigated, how was the activity and the resulting NAD⁺ levels affected? In other words, how potent is the KO in terms of affecting NAD⁺?

Response: In response to the reviewer's suggestion, we measured the NAD⁺ content in astrocytes obtained from the MBH of both WT and AKO mice. The resulting data are now shown in Figure 2c. The NAD⁺ content in MBH astrocytes from AKO mice exhibited a 50% reduction compared to the levels observed in WT mice.

- Fig 2D: Were control groups taken out to 16 weeks as well? How significant is the KO weight loss physiologically?

Response: Based on Reviewer #4's feedback, we have now presented the body weight data for the 12-week HFD feeding in the revised Figure 2d. Despite consuming an equal amount of calories, the body weights of AKO mice were 8%–10% lower than those of their WT littermates upon chronic HFD challenge. Furthermore, the fat mass in AKO mice was, on average, 22% lower than that in WT mice (Figures 1d, e). It is noteworthy that typical anti-obesity medications for treating human obesity usually target more than a 5% weight loss. Therefore, while the effect of weight loss due to astrocyte *Nampt* depletion may not be substantial, it is still physiologically accountable as mentioned in the Discussion (Line 350–352).

- Fig 2K/L: Along the same point, if the KO mice didn't improve insulin sensitivity or glucose tolerance, can the authors comment on the physiological relevance of the weight loss seen? Perhaps the HFD-mice were not taken out long enough to see an impairment in glucose tolerance/insulin resistance in the WT mice, and as such, no benefit could have been seen in the KO mice.

Response: We have now presented the GTT and ITT data in Figure 2k and l; these data were acquired from mice fed a HFD for 8 weeks. Upon analysis, we observed no significant differences in the GTT and ITT responses between AKO and WT mice following 8 weeks of HFD feeding. These findings suggest a potential dissociation in the regulation of energy and glucose metabolism mediated by astrocyte *Nampt*–NAD⁺ signaling. We have discussed this point in the Discussion section (Line 360–364).

- Fig 3D: typo in legend (*STAT3* not *START3*). Did authors observe differences in leptin receptor levels as well? A mechanism for this increased *STAT3* signaling upon *NAMPT* deletion would be useful to explore/discuss.

Response: Thank you for highlighting our mistake. We corrected this typo in Figure 3d. To explore the mechanism of altered leptin sensitivity, we measured the mRNA expression of the leptin receptor *Leprb* and the leptin signaling inhibitors *Socs3* and *Ptp1b* using qPCR analysis. AKO mice exhibited no significant alteration in

hypothalamic *Leprb* expression but showed a significant reduction in *Ptp1b* and *Socs3* expression, as depicted in Figure 3e. Therefore, the reduced expression of leptin signaling inhibitors may underlie the improved leptin signaling in HFD-fed AKO mice.

- Fig 3H: *In both neurons and microglia, the ACM-NAMPT O/E increases pro inflammatory cytokines that aren't all completely blocked by the neutralizing antibody. This could be due to the potency of the neutralizing antibody OR the fact that the effects are not all directly attributable to eNAMPT. To address this, looking at NAD+ content in these experiments would be beneficial, if feasible. Alternatively, authors can discuss the potency of the antibody if known from other studies.*

Response: To distinguish between two possibilities, we conducted tests to determine whether Nampt antibody cotreatment could effectively reverse the effects of *Nampt*-O/E ACM on microglia and neurons, as assessed by measuring the NAD⁺ levels in both neurons and microglia. As demonstrated in Supplementary Figure 5c, cotreatment with the NAMPT antibody affectively inhibited *Nampt*-O/E ACM-induced increases in cellular NAD⁺ levels in both neurons and microglia. These results confirmed the efficacy of NAMPT neutralizing antibody (Lines 242–249).

- Fig 4D: *What happens to CD38 activity in isolated MBH astrocytes from NAMPT KO animals fed a HFD (i.e samples from exp in 2D, only WT is shown in 4D)? Wouldn't this further reinforce that NAMPT-NAD+ acts via CD38?*

Response: As per your suggestion, we measured the CD38 activity in isolated MBH astrocytes from both WT and AKO mice. The results are now illustrated in Figure 4d, revealing that the CD38 activity in MBH astrocytes from AKO mice was 30% lower than that observed in those from WT mice (Lines 269–270).

- Fig 4G: *Comment on why the body weight gain of these mice is less than the ones in 2D on just the HFD alone?*

Response: In the 78c study, mice were equipped with osmotic pumps and ICV cannula, which could impact weight gain during HFD feeding. We commented on this in Lines 280–281.

- Discussion: *“In line with this, activation of the NAD⁺ salvage pathway through Nampt overexpression or palmitate treatment stimulates proinflammatory gene expression and TNF- α secretion in hypothalamic astrocytes.” From Fig 3F - palmitate has inflammatory actions that are not mediated by the NAMPT pathway as FK866 does not completely block the palmitate-mediated increase in IL6 and MCP-1. Authors should acknowledge this.*

Response: As you pointed out, the activation of the NAD⁺ salvage pathway through *Nampt* overexpression or palmitate treatment stimulates the expression of proinflammatory genes and TNF- α secretion in hypothalamic astrocytes. However, *Nampt* inhibition with FK866 only partially reversed the effect of palmitate treatment on *Il-6* and *Mcp1* expression, while FK866 cotreatment almost completely reversed the effect of *Nampt* overexpression on these genes. These data suggest that palmitate upregulates *Il-6* and *Mcp1* expression via both *Nampt*-dependent and -independent pathways, such as TLR4–NF κ B signaling and inflammasome activation. We discussed this aspect in the Result section (Lines 224–227).

- Overall this is a nice study that would benefit from some important controls in order to gain confidence in the results.

Thank you again for your detailed review of our study and helpful comments. We hope that our responses and the

corresponding revisions are satisfactory.

Reviewer #2 (Remarks to the Author):

Overall the study is well done, presented and discussed. We have the few comments mentioned below:

1. An assay directly measuring changes in CD38 levels or activity in hypothalamic astrocytes when NAMPT is manipulated (either knocked out with the hGFAP-Cre ERT2 or overexpressed in the Nampt O/E astrocytes) would better support the conclusion that it is acting downstream of the NAD⁺ salvage pathway, as stated in the abstract.

Response: As suggested, we measured the CD38 activity in isolated MBH astrocytes from AKO mice. As depicted in Figure 4d, the CD38 activity in hypothalamic astrocytes was 30% lower in AKO mice than in WT mice. We also measured CD38 activity in *Nampt* O/E astrocytes, as presented in Figure 4c, and observed an increase in astrocytic CD38 activity due to *Nampt* O/E. These findings support the notion outlined in the abstract that astrocyte CD38 acts as a downstream mediator of *Nampt*.

2. Measuring of hypothalamic astrocyte NAD⁺ levels in the AKO mouse would rule out any compensatory increase in press-handler and de-novo synthesis pathways that can produce NAD⁺ and that may be occurring an in-vivo setting.

Response: We measured the NAD⁺ content in MBH astrocytes from WT and AKO mice. The data are now presented in Figure 2c (Lines 130–131), revealing that the NAD⁺ content in MBH astrocytes from AKO mice was 50% lower than that in their WT littermates.

3. It would be better to match the mice for the leptin treatment experiment in Figure 3D for adiposity (fat mass) rather than just body weight (Supp figure 2A) as this would be a better control for the potential confounding effects of this parameter in leptin sensitivity.

Response: We appreciate the reviewer for helpful comment. We have included the fat mass data together with the body weight in Supplementary Figure 5a.

Thank you again for your detailed review of our study and helpful comments. We hope that our responses and the corresponding revisions are satisfactory.

Reviewer #3 (Remarks to the Author):

Reviewer #4 (Remarks to the Author):

The present work addresses the timely topic of high fat diet induced astrocyte reactivity and inflammation in the hypothalamus. Employing a combination of in vitro and in vivo techniques, the authors shed light on the role of the NAD⁺ salvage pathway and its downstream component, CD38, in relation to obesity-related metabolic aspects.

In their initial investigation, utilizing primary cultured hypothalamic astrocytes, the authors employed pharmacological inhibitors to convincingly establish the critical importance of the salvage pathway in driving the increase in NAD⁺ levels induced by palmitate in astrocytes. Subsequently, they generated mice with astrocyte-specific NAMPT depletion to target the NAD⁺ pathway and assess its implications in vivo. This experiment revealed that in the absence of an activated NAD⁺ salvage pathway, mice subjected to a high-fat diet exhibited reduced adipose tissue accumulation and displayed enhanced energy expenditure, despite having a similar caloric intake. These changes were associated with a decreased level of inflammatory processes in the arcuate nucleus, and these findings were corroborated using an alternative model involving shRNA injection in the arcuate nucleus. Furthermore, an examination of downstream NAD⁺-dependent enzymes in primary astrocyte cultures revealed that inhibiting SIRT1/2, did not alter the proinflammatory effects associated with the overexpression of Nampt. On the contrary, CD38blockade recapitulated the beneficial effects of NAMPT depletion in high fat diet fed animals.

Well controlled and synergistic approaches for targeting key components of the NAD⁺ pathway have been developed using transgenic mice, viral tools and pharmacology in vivo combined with metabolic assessments.

The manuscript is clearly written and organized. Here are my comments.

Majors:

1. One limitation of this study is to generalize the results derived from cultured astrocytes or from the large region of the mediobasal hypothalamus (MBH) to the entire hypothalamus. This limitation arises from the fact that astrocytes within different nuclei of the hypothalamus can exhibit significant functional variability and distinct responses when exposed to a high-fat diet (HFD). My recommendation is that the authors discuss this important limitation.

Response: We discussed the possibility of functional variability in astrocytes across distinct hypothalamic nuclei, acknowledging that the methods utilized in our study did not test this possibility in the Discussion section (Lines 344–347).

2. Page 24: regarding the section on hypothalamic AAV injections. In their approach to target the ASH, the authors inject a substantial volume of 500 µl of virus on each side that could lead to a larger infection than just the ARH. Could the author provide a figure showing the extent of the viral infection?

Response: We have provided the data showing that AAV infection was restricted to the ARH in Supplementary Figure 4a.

3. What was the rationale behind the authors selecting a 4-week exposure to a high-fat diet (HFD) in their experimental design? It would be insightful to understand whether it is based on metabolic considerations.

Response: In fact, we monitored the body weights up to 26 weeks of age (12 weeks of HFD feeding) but only presented the data for up to 4 weeks of HFD feeding, as the difference between groups remained similar during the extended period of observation. We have now included the body weight data for the entire monitoring period in the revised Figure 2d.

4. Even if the article is centered on astrocytes, it is interesting to notice the differential effect of palmitate on neurons and astrocytes since most of the literature is on NAD⁺ pathway in neurons. In Figure 1, it is observed that neurons exhibit a more rapid increase in NAD⁺ levels compared to astrocytes. This temporal distinction in NAD⁺ dynamics within these two cell types raises the question of the crosstalk and signaling between neurons and astrocytes. The interplay between neurons and astrocytes might impact NAD⁺ dynamics. All the culture experiments are astrocytes only. Could the authors discuss this point?

Response: As previous studies have focused on the hypothalamic neuronal NAD⁺ pathway, we chose to concentrate on the NAD⁺ pathway in hypothalamic astrocytes. Therefore, we did not perform all in vitro studies in hypothalamic neuronal cells, except for the study involving astrocyte conditioned medium. In response to an important point raised by the reviewer, we discussed the potential interplay between the NAD⁺ pathways of neurons and astrocytes, especially under chronic HFD-fed conditions (Lines 382–386).

5. Line 146, the authors mentioned that the effect was absent in females AKO mice. Did the authors tested females on the different group of mice studied? This is not mentioned in the methods.

Response: We evaluated body weight change, food intake, GTT, and ITT in WT and AKO female mice, but found no significant differences. These data are presented in Supplementary Figure 3. We mentioned this evaluation in the Methods (Lines 149–150, 464).

6. It would be beneficial to provide a more detailed description of the metabolic experiments of the study. Figures displays 2 days in the metabolic cages, and these 2 days appears quite different for EE (figure 2h). How long did the run last? Are the traces averaged over multiple days or not? Were the statistics performed on 2 days with dark and light phases together?

Response: We analyzed mice using the CLAMS over a 3-day period; the first day served as an adaptation period for the mice, while we used the data from the second and third days for analysis. We calculated the average EE (day and night) on the second and third days and used them for ANCOVA. We included these details in the Method (Lines 467–470).

7. In figure 4g and h, body weight gain and fat mass are decreased in 78c injected mice, could the authors provide the graph of the lean mass in h also? In figure 4h, is the regression significant for the 78c injected group?

Response: Mice subjected to ICV 78c treatment exhibited a reduction in lean mass, measured in grams; however, when expressed as a percentage of body weight, it remained unaltered. These findings are presented in Figure 4i.

8. In this specific group of mice, the authors have not supplied information regarding the locomotor activity. Could the variations in locomotor activity in the 78c mice be a contributing factor to the elevated energy expenditure in these particular mice?

Response: We presented the data relating to locomotor activity in Supplementary Fig. 7c, together with a description in Line 282. The locomotor activity did not differ significantly between the control and 78c groups, and thus did not contribute to the increased energy expenditure induced by 78c.

9. The rationale of the latest experiment is not fully clear to me. The authors demonstrate that leptin, insulin, and GLP-1 induce a calcium increase in hypothalamic astrocytes, which is subsequently affected by exposure to palmitate. Nevertheless, it's crucial to consider that various factors beyond the NAD⁺ salvage pathway, such as

receptor availability, could potentially contribute to this observed effect.

Response: Following the reviewer's comment, we discussed various factors that may affect calcium signaling in hypothalamic astrocytes in the Discussion section (Lines 418–422).

Minor comments :

In the methods section Alexa-Fluor is misspelled as Alexa-Flour

Line 327: "it has been shown" instead of "it have been shown".

Line 587: 78C the c is missing. In the same paragraph, there are multiple spelling mistakes

Response: Thank you for highlighting our mistakes. We have corrected all of the grammatical errors and typos.

Thank you again for your detailed review of our study and helpful comments. We hope that our responses and the corresponding revisions are satisfactory.

REVIEWERS' COMMENTS

Reviewer #1 (Remarks to the Author):

The authors have adequately addressed the majority of my comments. However, two minor comments remain:

In Fig 2k/l - authors state that there was a lack of improvement in the GTT in the KOs. But, was there an impairment in GTT/ITT to begin with? It would be important to clarify whether there was an impairment in GTT compared to regular CHOW fed mice which isn't clear.

Typo in new Fig 2d: Axis label still says body weight gain. It should be body weight (g).

Reviewer #3 (Remarks to the Author):

Reviewer #4 (Remarks to the Author):

The authors of the current manuscript have incorporated all essential elements that effectively address the concerns and inquiries raised during the review process.

Reviewer #5 (Remarks to the Author):

Reviewer's comments

Reviewer #1 (Remarks to the Author):

The authors have adequately addressed the majority of my comments. However, two minor comments remain:

In Fig 2k/l - authors state that there was a lack of improvement in the GTT in the KOs. But, was there an impairment in GTT/ITT to begin with? It would be important to clarify whether there was an impairment in GTT compared to regular CHOW fed mice which isn't clear.

Response: We have now included the data for GTT and ITT in Nampt-KO and Nampt-WT mice on a chow diet in Supplementary Fig. 2g, h and manuscript (Line 149). We have also demonstrated that GTT and ITT were impaired in HFD-fed condition compared to chow-diet-fed condition as presented below.

Typo in new Fig 2d: Axis label still says body weight gain. It should be body weight (g).

Response: Thank you for pointing out the typo in Fig. 2d. We have rectified this error.